# GNNBoundary: Towards Explaining Graph Neural Networks through the Lens of Decision Boundaries

**Xiaoqi Wang**
The Ohio State University
`wang.5502@osu.edu`

**Han-Wei Shen**
The Ohio State University
`shen.94@osu.edu`

## Abstract

While Graph Neural Networks (GNNs) have achieved remarkable performance on various machine learning tasks on graph data, they also raised questions regarding their transparency and interpretability. Recently, there have been extensive research efforts to explain the decision-making process of GNNs. These efforts often focus on explaining why a certain prediction is made for a particular instance, or what discriminative features the GNNs try to detect for each class. However, to the best of our knowledge, there is no existing study on understanding the decision boundaries of GNNs, even though the decision-making process of GNNs is directly determined by the decision boundaries. To bridge this research gap, we propose a model-level explainability method called GNNBoundary, which attempts to gain deeper insights into the decision boundaries of graph classifiers. Specifically, we first develop an algorithm to identify the pairs of classes whose decision regions are adjacent. For an adjacent class pair, the near-boundary graphs between them are effectively generated by optimizing a novel objective function specifically designed for boundary graph generation. Thus, by analyzing the near-boundary graphs, the important characteristics of decision boundaries can be uncovered. To evaluate the efficacy of GNNBoundary, we conduct experiments on both synthetic and public real-world datasets. The results demonstrate that, via the analysis of faithful near-boundary graphs generated by GNNBoundary, we can thoroughly assess the robustness and generalizability of the explained GNNs. The official implementation can be found at this GitHub repository.

## 1 Introduction

Graph Neural Networks (GNNs) are widely recognized as powerful tools for modeling graph data across various domains, including chemistry, social networks, transportation, etc (Lin et al., 2022). However, their success has raised questions regarding their transparency and interpretability, as GNNs are often regarded as black-box models, similar to other types of deep learning models (Wu et al., 2022). The complexity of GNNs makes it challenging for humans to comprehend their decision-making processes, posing a significant obstacle to their application in real-world problems (Chen et al., 2023b). For example, Partin et al. (2023) utilized GNN to predict cancer response (sensitive or resistant) to drug treatments. Nonetheless, if the decision-making process of this GNN is not adequately verified by humans, we cannot fully trust its prediction in real-world scenarios, especially when it comes to serious matters such as human lives. This underscores the necessity for developing methods to ensure the explainability of GNNs, enabling humans to gain insights into their inner working mechanisms and build trust in their decisions.

Over the past few years, there has been a growing interest in explaining GNNs, leading to extensive research efforts. These studies can be broadly categorized into two main groups: instance-level explanations (Ying et al., 2019; Luo et al., 2020) and model-level explanations (Yuan et al., 2020; Wang & Shen, 2023). Instance-level explanations are designed to explain why a certain prediction is being made for a specific instance, while model-level explanations aim to disclose high-level decision-making processes without respect to any particular instances. In this paper, the proposed

method can also be categorized as a model-level explanation. However, unlike the existing model-level explanation methods, we attempt to open the black box of GNNs from a different perspective.

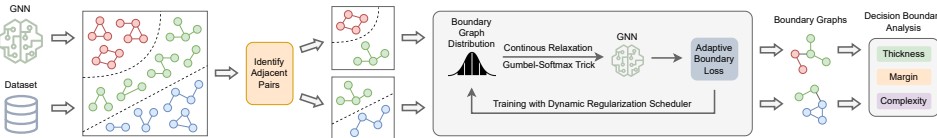

Figure 1: The overview of GNNBoundary. Given a trained GNN, GNNBoundary first identifies adjacent class pairs, and then generates boundary graphs for each pair. The generated boundary graphs can facilitate a deeper understanding of the decision boundaries of GNN.

In general, existing model-level methods explain the GNNs by generating a feature descriptor for each class in the graph classification task (Chen et al., 2023a). By analyzing the feature descriptors for each class they generate, one can understand the decision-making rule of the GNNs for individual classes (Wang & Shen, 2023). However, they often fail to answer the following questions when explaining the GNNs: (i) which classes are more similar to each other such that the explained GNN will easily get confused? (ii) in what circumstances will the GNN be more uncertain and less confident about its predictions? (iii) how to obtain a comprehensive overview of its decision-making schema that effectively captures the complex relationships between different classes? All of these questions are imperative for understanding the internal mechanism of GNNs, which strongly motivated us to get a deeper understanding of their decision boundary rather than solely focusing on individual classes. We believe that analyzing the decision boundary of GNNs is an indispensable step to fully understand their decision-making process, and we are taking the first step in this direction.

In this paper, we propose a model-level explainability method, GNNBoundary, to explain the decision boundaries of GNNs on graph classification tasks. Specifically, we first introduce an algorithm to identify pairs of classes whose decision regions are adjacent, given the fact that the decision boundaries would only exist between those class pairs. For each adjacent class pair, GNNBoundary generates graphs that are extremely close to the decision boundary by optimizing a novel objective function specifically designed for satisfying the two desired properties we propose for boundary graph generation. To facilitate the optimization process, a dynamic regularization scheduler is introduced to reduce the likelihood of trapping in the local minima. In the experimental study, we quantitatively and qualitatively evaluated the efficiency and effectiveness of GNNBoundary on both synthetic and public real-world datasets. The experimental results show that we could consistently generate faithful near-boundary graphs for each adjacent class pair. More importantly, we have demonstrated in the case studies that the generated boundary graphs can be utilized to measure the boundary thickness, boundary margin, and boundary complexity of GNNs trained on three datasets. These three metrics would allow us to gain deep insights into the complexity of decision boundaries, the potential risk of misclassification on unseen data, and the model robustness to perturbation.

## 2 RELATED WORK

**Instance-Level Explanations of GNNs.** Instance-level explanation methods provide input-dependent explanations for the explained GNNs. Their main goal is to explain why a certain prediction is made for a particular instance. Specifically, the important input features that contribute the most to model predictions are identified for a given instance. Based on a recent survey (Kakkad et al., 2023), instance-level explanation methods can be decomposed into five different categories: decomposition methods (Schnake et al., 2021), gradient-based methods (Pope et al., 2019), surrogate methods (Duval & Malliaros, 2021), perturbation-based methods (Ying et al., 2019), and generation-based methods (Luo et al., 2020). They determine the important input features for the prediction of a particular instance in a different way.

**Model-Level Explanations of GNNs.** On the contrary, model-level explanation methods focus on explaining the high-level decision-making process of GNNs, which is not specific to any particular instance. However, this is much less explored than instance-level methods. Generally speaking, the existing model-level methods often produce a feature descriptor to describe the most discriminative features GNNs try to detect for each class. This feature descriptor could be in the form of

an explanation graph (Wang & Shen, 2023; Yuan et al., 2020) or a logic formula (Azzolin et al., 2023). GNNBoundary also fall into this category. Similar to GNNInterpreter (Wang & Shen, 2023), GNNBoundary also adopts the Gumbel-Softmax trick (Jang et al., 2016) to generate a graph that minimizes an objective function. However, there is a fundamental difference in the objective function: GNNBoundary attempts to generate the graphs lying on decision boundaries, whereas GNNInterpreter aims at generating graphs with the most discriminative pattern of individual classes. It is worth mentioning that, to the best of our knowledge, GNNBoundary is the first explainability method that focuses on explaining and analyzing the decision boundaries of GNNs.

**Explaining Decision Boundary of Neural Networks.** Although there are no previous studies on explaining the decision boundaries of GNNs, there are few studies on analyzing the decision boundaries of Convolutional Neural Networks (CNNs) or Multilayer Perceptron (MLP). In summary, they mainly focus on analyzing model robustness (Yang et al., 2020), the dynamic of margin during training (Mickisch et al., 2020), and model generalizability (Guan & Loew, 2020). One common approach is generating near-boundaries instances and analyzing those instances to gain insights about the characteristics of decision boundaries (Berk et al., 2022). Inspired by this idea, we also aim to generate boundary graphs with the purpose of understanding the decision boundaries of GNNs. Given those boundary graphs, we have demonstrated in the case studies that the robustness and generalizability of GNNs could be adequately assessed.

## 3 BACKGROUND

**Notations.** A graph is denoted by $G = (\mathcal{V}, \mathcal{E})$, where $\mathcal{V} = \{v_1, v_2...v_N\}$ is the node set and $\mathcal{E} \subseteq \mathcal{V} \times \mathcal{V}$ is the edge set. $M$ and $N$ represent the number of edges and nodes, respectively. The node adjacency information is encoded by an adjacency matrix $\mathbf{A} \in \{0, 1\}^{N \times N}$, where $a_{ij} = 1$ when there exists an edge connecting node $v_i$ and node $v_j$, and $a_{ij} = 0$ otherwise. Besides, the node attributes are represented by the node feature matrix $\mathbf{Z} \in \mathbb{R}^{N \times d}$. For a graph neural network $f$ with $L$ layers, $\phi_l$ denotes the embedding function composed of the first $l$ layers of the network which takes as input a graph $G$, while $\eta_l$ denotes the scoring function that corresponds to the last $L - l$ layers of the network $f$ before softmax.

**Graph Neural Networks.** The general idea of GNNs is the message-passing schema, which can be decomposed into three operations: computing messages, aggregating messages, and updating the hidden node representations (Wang et al., 2021). For each GNN layer $l$, these three operations will be performed sequentially to produce the hidden node representations $\mathbf{H}^{(l)} = \phi_l(G)$ where $\mathbf{H}^{(l)} \in \mathbb{R}^{N \times D^{(l)}}$. Specifically, at the computing messages step, the messages $\mathbf{m}_{ij}^{(l)} = \text{Compute}(\mathbf{h}_i^{(l-1)}, \mathbf{h}_j^{(l-1)})$ are computed based on the hidden node representation of node $v_i$ and $v_j$ at the previous layer. Then, the messages propagated from the neighboring nodes are aggregated with a function $\mathbf{m}_i^{(l)} = \text{Aggregate}(\{\mathbf{m}_{ij}^{(l)} | v_j \in \mathcal{N}_{v_i}\})$. Lastly, the output hidden representation at the current layer is updated via $\mathbf{h}_i^{(l)} = \text{Update}(\mathbf{m}_i^{(l)}, \mathbf{h}_i^{(l-1)})$, based on the aggregated message and the hidden representation of itself at the previous layer.

**Decision Region and Decision Boundary.** Given a graph classifier $f$ with $L$ layers, it partitions the input graph space and each embedding space into $C$ decision regions $\{\mathcal{R}_c^{(l)} | c \in [1, C]\}$ for each layer $l$, where we denote $\mathcal{R}_c = \mathcal{R}_c^{(0)}$ as the decision region for the input space. For each graph $G \in \mathcal{R}_c$, the graph classifier $f$ predicts $G$ as class $c$, where $c = \text{argmax}_k f_k(G)$. Besides, the decision boundary between class $c_1$ and class $c_2$ is defined as $\mathcal{B}_{c_1 \| c_2} = \{G : f_{c_1}(G) = f_{c_2}(G) > f_{c'}(G), \forall c' \neq c_1, c_2\}$, when the graph classifier believes that the graph $G$ has an equal probability of belonging to class $c_1$ and $c_2$. For the embedding spaces, we define $\mathcal{B}_{c_1 \| c_2}^{(l)} = \{\mathbf{H}^{(l)} : \sigma(\eta_l(\mathbf{H}^{(l)}))_{c_1} = \sigma(\eta_l(\mathbf{H}^{(l)}))_{c_2} > \sigma(\eta_l(\mathbf{H}^{(l)}))_{c'}, \forall c' \neq c_1, c_2\}$.

## 4 GNNBOUNDARY

We propose a model-level explainability method, GNNBoundary, to explain the high-level decision-making process of GNNs from the perspective of decision boundaries. We first develop an algorithm to identify the pairs of classes that have adjacent decision regions. Then, we design a novel objective

function that can effectively generate faithful near-boundary graphs for adjacent class pairs via the Gumbel-Softmax trick (Jang et al., 2016). Besides, to facilitate the optimization, a dynamic regularization scheduler is introduced to reduce the likelihood of being trapped in the local minima. Thus, the generated near-boundary graphs can be used to analyze the robustness and generalizability of GNNs. A high-level overview of GNNBoundary is shown in Figure 1.

## 4.1 Identifying Adjacent Classes

In order to find a boundary graph $G_{c_1\|c_2} \in \mathcal{B}_{c_1\|c_2}$, it is important to know how likely $G_{c_1\|c_2}$ actually exists. However, this information can not be directly obtained from the input graph space. Instead, it can be estimated from the embedding space of $f$, because $G_{c_1\|c_2}$ exists only if boundary embeddings $\mathbf{H}_{c_1\|c_2}^{(l)} \in \mathcal{B}_{c_1\|c_2}^{(l)}$ exist in the embedding space. We can determine the likelihood that $G_{c_1\|c_2}$ exists by measuring how ubiquitously $\mathbf{H}_{c_1\|c_2}^{(l)}$ exists in between $\mathcal{R}_{c_1}^{(l)}$ and $\mathcal{R}_{c_2}^{(l)}$. We define the ubiquity of $\mathbf{H}_{c_1\|c_2}^{(l)}$ as the probability that $\mathbf{H}_{c_1\|c_2}^{(l)}$ exists in between a pair of embeddings $\mathbf{H}_{c_1}^{(l)} = \phi_l(G_{c_1}) \in \mathcal{R}_{c_1}^{(l)}$ and $\mathbf{H}_{c_2}^{(l)} = \phi_l(G_{c_2}) \in \mathcal{R}_{c_2}^{(l)}$. For simplicity, we choose to use the embedding space of the last hidden layer output, where the decision boundaries are linear. Thus, we define the degree of adjacency of two classes $c_1$ and $c_2$ as the ubiquity of $\mathbf{H}_{c_1\|c_2}^{(L-1)}$, and $c_1$ and $c_2$ is said to be adjacent if their degree of adjacency is greater than a certain threshold.

To find the class pairs $c_1$ and $c_2$, where boundary graph $G_{c_1\|c_2}$ is more likely to exist, we develop an algorithm (shown in Algorithm 1) to measure the degree of adjacency between a pair of classes. Following the definitions above, we apply Monte Carlo to randomly sample $K$ pairs of graphs $G_{c_1} \in \mathcal{R}_{c_1}$ and $G_{c_2} \in \mathcal{R}_{c_2}$, then determine if $\mathbf{H}_{c_1\|c_2}^{(L-1)} \in \mathcal{B}_{c_1\|c_2}^{(L-1)}$ can be found by interpolating between $\mathbf{H}_{c_1}^{(L-1)}$ and $\mathbf{H}_{c_2}^{(L-1)}$. Due to the linearity of the decision boundary, this can be done by checking if we need to pass through any other decision region $\mathcal{R}_{c'}^{(L-1)}$ when walking from $\mathbf{H}_{c_1}^{(L-1)}$ to $\mathbf{H}_{c_2}^{(L-1)}$ in a straight line. As a result, the degree of adjacency between $c_1$ and $c_2$ can be derived by computing the ratio of $\mathbf{H}_{c_1\|c_2}^{(L-1)}$ found in $K$ sampled pairs. By choosing a proper threshold, the adjacent class pairs can be identified from the adjacency degree matrix for each pair of classes.

---

**Algorithm 1:** Measure the Degree of Adjacency of a Class Pair

---

1   count $\leftarrow 0$
2   **for** $k \leftarrow 1...K$ **do**
3      Randomly sample two graphs $G_{1_k} \in \mathcal{R}_{c_1}$ and $G_{2_k} \in \mathcal{R}_{c_2}$
4      Compute score $= \prod\limits_{\lambda \in [0,1]} \mathbb{1}_{\{c_1,c_2\}}(\arg\max\limits_{c} \eta_{L-1}(\lambda\phi_{L-1}(G_{1_k}) + (1-\lambda)\phi_{L-1}(G_{2_k})))$
5      **if** score $\neq 0$ **then**
6          count $\leftarrow$ count $+ 1$
7   **return** $\frac{\text{count}}{K}$

---

## 4.2 Learning Objective of Boundary Graphs Generation

To effectively generate boundary graphs, the learning objective needs to be properly defined to balance the trade-off between the two boundary classes. For the sake of effective and efficient optimization, we propose two desired properties of the objective function for the purpose of boundary graph generation. Suppose $c_1$ and $c_2$ is an adjacent class pair of our interest. (1) For $b \in \{c_1, c_2\}$, the objective function should encourage $p(b)$ if $p(b) < 0.5$ and discourage $p(b)$ otherwise. For $b' \notin \{c_1, c_2\}$, the objective function should always discourage posterior probability $p(b')$. (2) The objective function should always encourage logit values $f(G)_b$ for $b \in \{c_1, c_2\}$ and discourage $f(G)_{b'}$ for $b' \notin \{c_1, c_2\}$. As suggested by Berk et al. (2022), the cross-entropy loss below seems to naturally fit our purpose and also satisfy the desired property (1),

$$\min_G \mathcal{L}(G) = \min_G - \sum_{c \in [1,C]} (\mathbf{p}_{\text{target}})_c \log \sigma(f(G))_c \tag{1}$$

where $\sigma$ denotes the softmax function and $\mathbf{p}_{\text{target}}$ is a $C$-dimensional vector whose $c_1$-th and $c_2$-th element are both 0.5, and all other elements are 0. However, we can show that minimizing the cross-entropy loss above is not efficient enough for the specific purpose of generating boundary graphs, as it does not satisfy the desired property (2). We have proved the following proposition in Appendix A,

**Proposition 4.1.** *Minimizing the cross-entropy loss is equivalent to minimizing the function below,*

$$\min_G \sum_{b' \notin \{c_1, c_2\}} f(G)_{b'} \cdot p^*(b') - \sum_{b \in \{c_1, c_2\}} f(G)_b \cdot \big(0.5 - p^*(b)\big), \tag{2}$$

*where $p^*(c)$ is the detached version (no gradient) of the posterior probability $p(c) = \sigma(f(G))_c$.*

**Corollary 4.1.** *In certain cases, even if $b$ belongs to one of the boundary classes, namely $b \in \{c_1, c_2\}$, minimizing the cross-entropy loss would still discourage the logit value $f(G)_b$.*

From Corollary 4.1, it shows a clear limitation of cross-entropy loss that the logit value for either $c_1$ or $c_2$ can be minimized in some scenarios, which somehow contradicts our purpose of generating a boundary graph that is as similar as possible to both $c_1$ and $c_2$ in terms of the discriminative graph patterns. Therefore, we propose a novel objective function that encourages $p(c) = p(c) = 0.5$ while ensuring $f(G)_{c_1}$ and $f(G)_{c_2}$ are never minimized throughout training. The proposed objective function of generating boundary graphs, which satisfies the two desired properties, is written as

$$\min_G \mathcal{L}(G) = \min_G \sum_{b' \notin \{c_1, c_2\}} \beta f(G)_{b'} \cdot p^*(b')^2 - \sum_{b \in \{c_1, c_2\}} \alpha f(G)_b \cdot \big(1 - p^*(b)\big)^2 \cdot \mathbb{1}_{p^*(b) < \max_{c \in [1, C]} p^*(c)}, \tag{3}$$

where $\alpha$ and $\beta$ are constant hyper-parameters. In addition to satisfying the two aforementioned desired properties, comparing to Equation 2, the additional square on the logit weight terms, $\big(1 - p^*(b)\big)^2$ and $p^*(b')^2$, further promote or penalize logit values that deviate significantly from the target class probability vector $\mathbf{p}_{\text{target}}$. Based on our empirical observations in Appendix C, this adaptive loss function effectively generates near-boundary graphs with faster convergence and a reduced likelihood of being trapped in local minima.

### 4.3 LEARNING BOUNDARY GRAPHS DISTRIBUTION

Due to the discrete nature of the graph data, the objective function in Equation 3 cannot be directly optimized via gradient-based methods, because $\nabla_{\mathbf{A}} \mathcal{L}(G)$ does not exist. To resolve this issue, we continuously relax the graph $G$ and optimize the objective function via reparameterization trick (Jang et al., 2016), inspired by Wang & Shen (2023) and Luo et al. (2020). Adopting the reparameterization trick with continuous relaxation of graphs has the advantage that it can make our approach applicable to generating boundary graphs with discrete node features and discrete edge features efficiently and effectively. Here, we only present the mathematical formulation for generating boundary graphs with discrete node features but without edge features for simplicity.

Assuming the boundary graph is a Gilbert random graph (Gilbert, 1959) and the node features are independently distributed, we formulate the boundary graph distribution as follows,

$$P(G) = \prod_{v_i \in \mathcal{V}} P(z_i) \cdot \prod_{(v_i, v_j) \in \mathcal{E}} P(a_{ij}) \tag{4}$$

where $a_{ij} = 1$ if node $v_i$ and $v_j$ are connected and $a_{ij} = 0$ otherwise, and $z_i$ denotes a categorical node feature of node $v_i$. A straightforward instantiation of $P(a_{ij})$ and $P(z_i)$ are $a_{ij} \sim \text{Bernoulli}(\theta_{ij})$ and $z_i \sim \text{Categorical}(\mathbf{p}_i)$ with $\|\mathbf{p}_i\|_1 = 1$. To make the graph $G = (A, Z)$ differentiable with respect to our objective function, we continuously relax $a_{ij}$ to be $\tilde{a}_{ij} \in [0, 1]$ and $z_i$ to be $\tilde{\mathbf{z}}_i \in [0, 1]^d, \|\tilde{\mathbf{z}}_i\|_1 = 1$. Since the Concrete distribution (Maddison et al., 2016) is a continuous version of Categorical distribution with closed-form density, we have $\tilde{a}_{ij} \sim \text{BinaryConcrete}(\omega_{ij}, \tau_a)$ and $\tilde{\mathbf{z}}_i \sim \text{Concrete}(\zeta_i, \tau_z)$, where $\tau_a$ and $\tau_z$ are the hyper-parameters to control the approximation of Categorical distribution, $\omega_{ij} \in \mathbf{\Omega}$ and $\zeta_i \in \mathbf{\mathcal{Z}}$. To make the sampling procedure of $\tilde{a}_{ij}$ and $\tilde{\mathbf{z}}_i$ differentiable, the Gumbel-Softmax trick (Jang et al., 2016) is adopted to sample $\tilde{a}_{ij}$ and $\tilde{\mathbf{z}}_i$ as: $\tilde{a}_{ij} = \text{sigmoid}\left((\omega_{ij} + \log \epsilon - \log(1 - \epsilon))/\tau_a\right)$ and $\tilde{\mathbf{z}}_i = \text{Softmax}\left((\zeta_i - \log(-\log \epsilon))/\tau_z\right)$, where $\epsilon \sim \text{Uniform}(0, 1)$. Thanks to this trick, we can sample $\tilde{a}_{ij}$ and $\tilde{\mathbf{z}}_i$ as an approximation of $a_{ij}$ and $z_i$ in a differentiable sampling procedure. Thus, the distribution of boundary graphs can be learned by minimizing the following objective function via Monte Carlo and gradient descent,

$$\min_{\mathbf{A}, \mathbf{Z}} \mathcal{L}(G) = \min_{\mathbf{\Theta}, \mathbf{P}} \mathbb{E}_{G \sim P(G)}[\mathcal{L}(\mathbf{A}, \mathbf{Z})] \approx \min_{\mathbf{\Omega}, \mathbf{\mathcal{Z}}} \mathbb{E}_{\epsilon \sim U(0, 1)}[\mathcal{L}(\tilde{\mathbf{A}}, \tilde{\mathbf{Z}})] \approx \min_{\mathbf{\Omega}, \mathbf{\mathcal{Z}}} \frac{1}{K} \sum_{k=1} \mathcal{L}(\tilde{\mathbf{A}}, \tilde{\mathbf{Z}}). \tag{5}$$

## 4.4 Training GNNBoundary

**Near-Boundary Criterion.** The training procedure should terminate when it successfully generates a boundary graph. However, it is practically not possible to generate boundary graph $G_{c_1 \| c_2} \in \mathcal{B}_{c_1 \| c_2}$ with $\sigma(f(G))_{c_1} = \sigma(f(G))_{c_2} = 0.5$ exactly. Thus, a relaxed criterion is needed to determine whether a graph $G$ approximately belongs to $\mathcal{B}_{c_1 \| c_2}$. We define the near-boundary criterion as,

$$\Psi(G) = \mathbb{1}_{p(c_1), p(c_2) \in [p_{\min}, p_{\max}]}(G). \tag{6}$$

**Regularization.** In addition to the proposed objective function in Equation 3, we apply multiple regularization terms to facilitate the training process and impose desired constraints on the generated boundary graphs. Following GNNInterpreter (Wang & Shen, 2023), our regularization terms include $L_1$ and $L_2$ regularizations $R_{L_s}$ for $s \in \{1, 2\}$ that mitigates saturating gradient problem, as well as the budget penalty $R_{\text{budget}}$ that encourages the succinct boundary graph to be generated.

$$R_{\text{budget}} = \text{Softplus}\left(\|\text{sigmoid}(\mathbf{\Omega})\|_1 - B\right)^2, \tag{7}$$

where $B$ is the anticipated maximum number of edges in the boundary graph $G$.

**Dynamic Regularization Scheduler.** Due to the unique characteristics of the graph data optimization problem, where a budget penalty is employed to regulate the graph size, this additional constraint may potentially hinder convergence. To mitigate this side effect, we propose a dynamic scheduling method within the training procedure that adaptively adjusts the budget penalty weight. Intuitively speaking, we impose a smaller budget penalty weight when the input graph does not meet the near-boundary criterion and progressively increase the weight as it approaches the target. Formally, the budget penalty weight $w_{\text{budget}}^{(t)}$ for iteration $t$ is recursively defined as,

$$w_{\text{budget}}^{(t)} = w_{\text{budget}}^{(t-1)} \cdot s_{\text{inc}}^{\mathbb{1}\{\Psi(G^{(t)})\}} \cdot s_{\text{dec}}^{\mathbb{1}\{\neg\Psi(G^{(t)}) \wedge (s_{\text{dec}} \cdot w_{\text{budget}}^{(t-1)} \geq w_{\text{budget}}^{(0)})\}}, \tag{8}$$

where $G^{(t)} = \mathbb{E}_{G \sim P(G)}[G]$ for iteration $t$, and $w_{\text{budget}}^{(0)}$, $s_{\text{inc}}$, and $s_{\text{dec}}$ are hyper-parameters for initial weight, weight increment, and weight decrement, respectively. Compared with constant scheduling of budget penalty, dynamic scheduling prevents the budget penalty from interfering with the main loss function during the early stage of the optimization and thus facilitates the convergence of optimization. The overall training procedure for GNNBoundary is presented in Algorithm 2.

---

**Algorithm 2:** Training GNNBoundary with Dynamic Regularization Scheduler

---

1   Initialize sampler parameters $\mathbf{\Omega}$ and $\mathcal{Z}$
2   **for** *each iteration $t$* **do**
3      Sample a batch of $k$ graphs $\{G_1...G_K\}$ from sampler with parameters $\mathbf{\Omega}$ and $\mathcal{Z}$
4      loss $\leftarrow \frac{1}{K} \sum_k \mathcal{L}(G_k) + w_{\text{budget}}^{(t)} \cdot R_{\text{budget}}(\mathbf{\Omega}) + w_L \cdot R_{L_s}(\mathbf{\Omega}, \mathcal{Z})$
5      Minimize loss with respect to $\mathbf{\Omega}$ and $\mathcal{Z}$
6      **if** $\Psi(\mathbb{E}[G]) = 1$ *and the size of $\mathbb{E}[G] < B$* **then**
7         **return** $\mathbf{\Omega}$ and $\mathcal{Z}$

---

## 5 Evaluation and Case Study

To thoroughly assess the efficacy and effectiveness of GNNBoundary, we conduct experimental studies on both synthetic and real-world datasets. First, a detailed description of the datasets will be presented in Section 5.1. For these datasets, the adjacent class pairs and the faithfulness of generated boundary graphs will be evaluated and discussed in Section 5.2. Lastly, we will perform three case studies in Section 5.3 to showcase how the generated boundary graphs can assist in analyzing the decision boundaries. The additional experimental details can be found in Appendix B.

## 5.1 Datasets and Graph Classifiers

We evaluated GNNBoundary on one synthetic dataset and two real-world datasets. **Motif** is a synthetically generated dataset following the same generation procedure as GNNInterpreter (Wang &

Shen, 2023). It contains four classes: House, House-X, Complete-4, and Complete-5, each of which corresponds to a motif that would appear in the graphs. A graph classifier has been trained on its training set with 10,378 graphs and obtained a test accuracy of 0.99 on 1,153 test graphs. **Collab** (Yanardag & Vishwanathan, 2015) is a real-world dataset related to the scientific Collaboration network in multiple different fields. The classes in this dataset represent ego networks of researchers in three scientific fields in physics: High Energy (HE), Condensed Matter (CM), and Astro. Collab contains 4,500 training graphs and 500 test graphs, and the trained graph classifier achieves a test accuracy of 0.74. **Enzymes** (Borgwardt et al., 2005) is a real-world molecule dataset with 6 classes and 3 types of nodes. Each class corresponds to one type of enzyme. The graph classifier to be explained is trained on 540 graphs and tested on 60 graphs with a test accuracy of 0.52.

## 5.2 EVALUATION RESULTS

On these three datasets, we carefully analyzed the degree of adjacency between every pair of classes and quantitatively evaluated the near-boundary graphs generated by GNNBoundary. Unfortunately, since there is no existing work on understanding the decision boundaries of GNNs, we have no existing approach to compare with. However, to better assess the effectiveness of GNNBoundary, we created a baseline method that generates boundary graphs by connecting a randomly sampled pair of graphs, $G_1 \in \mathcal{R}_{c_1}$, $G_2 \in \mathcal{R}_{c_2}$, with a random edge. It follows a general idea that the boundary graphs should contain the discriminative features of both $c_1$ and $c_2$. Due to the page limit, the quantitative comparison with another baseline approach using cross-entropy loss and the qualitative comparison with GNNInterpreter are presented in Appendix C and Appendix D, respectively.

**Adjacency between Classes.** First, it is essential to identify pairs of classes with adjacent decision regions via Algorithm 1. It is worth noting that the sampled graph pairs, $G_{1,k} \in \mathcal{R}_{c_1}$ and $G_{2,k} \in \mathcal{R}_{c_2}$, could be either the true graphs from the training data or the synthetically generated graphs. In this experimental study, we choose to use the synthetic graphs generated by GNNInterpreter because we do not want to make assumptions about having full access to the training data of the GNNs being explained. With the threshold value of 0.8, we identified 3, 2, and 6 adjacent class pairs, for the Motif, Collab, and Enzymes datasets, respectively (see Figure 2). For the Motif dataset, it is interesting to see that three non-adjacent pairs indeed are easier to differentiate compared with 3 adjacent pairs, based on the true motif presented in Figure 4. For example, regarding Complete-5 vs. House, Complete-5 has much more edge connectivity than House.

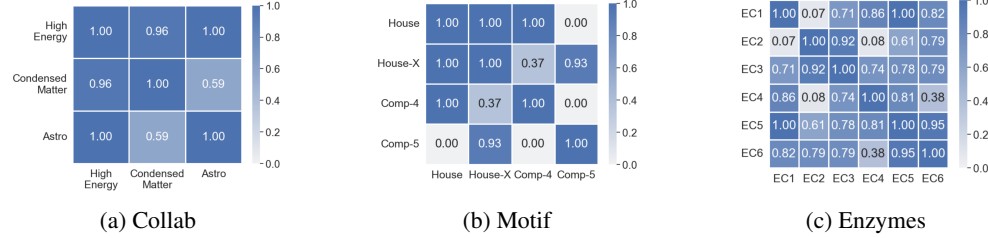

(a) Collab  (b) Motif  (c) Enzymes

Figure 2: The degree of adjacency between every pair of classes. It is computed with 10,000 randomly sampled pairs of graphs generated by GNNInterpreter.

**Generated Boundary Graphs.** To demonstrate the effectiveness of GNNBoundary on generating faithful boundary graphs, we generated 500 near-boundary graphs for each adjacent class pair, given the near-boundary criterion $p_{\min} = 0.45$ and $p_{\max} = 0.55$. Based on Table 1, it is evident that the average predicted class probability of those generated boundary graphs are all centered around 0.5 with a very small standard deviation, for all GNNs trained on three datasets. This result demonstrates that we can consistently generate boundary graphs that are extremely close to the decision boundary of interest. However, the boundary graphs generated by the random baseline approach obtain an average prediction probability far away from 0.5 for all three datasets, even though each generated graph combines a pair of training graphs from both $c_1$ and $c_2$. It is clear that GNNBoundary significantly outperforms the baseline approach on all three datasets.

Table 1: The quantitative evaluation of boundary graphs generated by both GNNBoundary and the baseline approach. The average predicted class probability of 500 generated boundary graphs along with the corresponding standard deviation is presented below. Besides, the complexity (Equation 11) is computed with the boundary graphs generated by GNNBoundary, ranging from 0 to 1.

| Dataset | $c_1$ | $c_2$ | GNNBoundary | | | Baseline | |
| --- | --- | --- | --- | --- | --- | --- | --- |
| | | | Complexity | $p(c_1)$ | $p(c_2)$ | $p(c_1)$ | $p(c_2)$ |
| Motif | House | HouseX | 6.55e-8 | $0.501 \pm 0.028$ | $0.499 \pm 0.028$ | $0.041 \pm 0.136$ | $0.949 \pm 0.145$ |
| | House | Comp4 | 6.55e-8 | $0.498 \pm 0.028$ | $0.501 \pm 0.028$ | $0.279 \pm 0.278$ | $0.717 \pm 0.285$ |
| | HouseX | Comp5 | 2.57e-7 | $0.491 \pm 0.026$ | $0.509 \pm 0.026$ | $0.007 \pm 0.049$ | $0.993 \pm 0.049$ |
| Collab | HE | CM | 0.0463 | $0.473 \pm 0.015$ | $0.487 \pm 0.016$ | $0.835 \pm 0.327$ | $0.153 \pm 0.318$ |
| | HE | Astro | 0.0246 | $0.526 \pm 0.013$ | $0.466 \pm 0.013$ | $0.364 \pm 0.462$ | $0.631 \pm 0.465$ |
| Enzymes | EC1 | EC4 | 0.0754 | $0.489 \pm 0.023$ | $0.487 \pm 0.021$ | $0.249 \pm 0.368$ | $0.219 \pm 0.365$ |
| | EC1 | EC5 | 0.1413 | $0.492 \pm 0.023$ | $0.489 \pm 0.025$ | $0.242 \pm 0.365$ | $0.360 \pm 0.430$ |
| | EC1 | EC6 | 0.1492 | $0.485 \pm 0.028$ | $0.472 \pm 0.017$ | $0.225 \pm 0.351$ | $0.245 \pm 0.358$ |
| | EC2 | EC3 | 0.2325 | $0.488 \pm 0.025$ | $0.488 \pm 0.025$ | $0.148 \pm 0.290$ | $0.239 \pm 0.363$ |
| | EC4 | EC5 | 0.1363 | $0.480 \pm 0.024$ | $0.486 \pm 0.024$ | $0.268 \pm 0.381$ | $0.351 \pm 0.419$ |
| | EC5 | EC6 | 0.2120 | $0.481 \pm 0.022$ | $0.486 \pm 0.023$ | $0.391 \pm 0.432$ | $0.269 \pm 0.368$ |

## 5.3 CASE STUDY

In these case studies, we utilized the generated boundary graphs, which served as an approximation to the decision boundary learned by GNNs, to measure the boundary thickness, boundary margin, and boundary complexity. Each of them provides unique insights into the decision boundaries of the explained GNNs from different perspectives. In these case studies, $G_c \in \mathcal{R}_c$ are all generated by GNNInterpreter rather than obtained from the training data. This decision is driven by two primary reasons: (i) we aim to avoid making assumptions about the accessibility of training data, and (ii) we hope to gain insights into the decision boundaries that extend beyond in-distribution data.

**Boundary Margin.** A classifier with a large margin can have better generalization properties and robustness to input perturbation (Elsayed et al., 2018). Formally, following Yang et al. (2020), the asymmetric margin of boundaries can be defined as,

$$\Phi(f, c_1, c_2) = \min_{(G_{c_1}, G_{c_1 \| c_2})} \|\phi_l(G_{c_1}) - \phi_l(G_{c_1 \| c_2})\| \tag{9}$$

where $(G_{c_1}, G_{c_1 \| c_2})$ could be any pair that $G_{c_1} \in \mathcal{R}_{c_1}$ and $G_{c_1 \| c_2} \in \mathcal{B}_{c_1 \| c_2}$, and $\phi_l$ is the graph embedding function of graph classifier $f$. As pointed out by Elsayed et al. (2018), the margin of deep networks can be defined based on any intermediate representation of the classifier and the ultimate decision boundary. In this paper, for the choice of $\phi_l$, we calculate the boundary margin based on the output embedding from the graph pooling layer, in line with the standard practices for interpreting deep neural networks (Bajaj et al., 2021). Given the graphs generated by GNNBoundary, the asymmetric boundary margin for each adjacent pair is presented in Figure 3, which demonstrates the inherent relationship between margin and robustness to perturbation. For the GNN trained on Motif, the margin from $\mathcal{R}_{\text{HouseX}}$ to $\mathcal{B}_{\text{House} \| \text{HouseX}}$ is 1.64 smaller than the margin to $\mathcal{B}_{\text{HouseX} \| \text{Comp5}}$, which would potentially result in higher risk of misclassifying House into Comp4 compared with HouseX. This is further verified by the confusion matrix in Figure 3. For the GNN trained on Collab, the margin from $\mathcal{R}_{\text{HE}}$ to $\mathcal{B}_{\text{HE} \| \text{Astro}}$ is smaller than the margin to $\mathcal{B}_{\text{HE} \| \text{CM}}$, which also expose a higher potential risk of misclassifying HE into Astro compared with CM (consistent with the confusion matrix). Similar observation can be obtained for the GNN on Enzymes, by comparing the margin from $\mathcal{R}_{\text{EC1}}$ to $\mathcal{B}_{\text{EC1} \| \text{EC5}}$ and that from $\mathcal{R}_{\text{EC1}}$ to $\mathcal{B}_{\text{EC1} \| \text{EC4}}$. Therefore, since the risk of misclassification is inherently related to the robustness of boundaries to perturbation, our case study shows that a larger margin may suggest greater robustness to perturbation over graphs.

**Boundary Thickness.** It has been shown that thick decision boundaries enhance the model robustness, whereas thin decision boundaries would result in overfitting and reduced robustness (Yang et al., 2020). Following Yang et al. (2020), the asymmetric boundary thickness is defined as follows,

$$\Theta(f, \gamma, c_1, c_2) = \mathbb{E}_{(G_{c_1}, G_{c_1 \| c_2}) \sim P} \left[ \|\phi_l(G_{c_1}) - \phi_l(G_{c_1 \| c_2})\| \int_0^1 \mathbb{1}_{\gamma > \sigma(\eta_l(h(t)))_{c_1} - \sigma(\eta_l(h(t)))_{c_2}} dt \right] \tag{10}$$

where $h(t) = (1 - t) \cdot \phi_l(G_{c_1}) + t \cdot \phi_l(G_{c_1 \| c_2})$ for $t \in [0, 1]$, $\phi_l$ is the graph embedding function of graph classifier $f$, $G_{c_1} \in \mathcal{R}_{c_1}$, and $G_{c_1 \| c_2} \in \mathcal{B}_{c_1 \| c_2}$. Similar to Yang et al. (2020), we let $\gamma = 0.75$ when computing the asymmetric boundary thickness in Figure 3. From analyzing the boundary

thickness matrix of GNNs trained on three datasets, we can identify their corresponding thickest and thinnest decision boundaries. For example, for the GNN trained on Collab, the thickest boundary is from $\mathcal{R}_{\text{HE}}$ to $\mathcal{B}_{\text{HE}\|\text{CM}}$ with a thickness of 11.52. This indicates that $G_{\text{HE}}$ might have greater adversarial robustness and out-of-distribution robustness with respect to Condensed Matter.

**Boundary Complexity.** The complexity of decision boundaries measures their generalization ability, which is crucial in determining how effectively the GNNs will perform on unseen graphs. Similar to Guan & Loew (2020), the complexity measure ranging from 0 to 1 can be computed by

$$\Gamma(f, c_1, c_2) = H(\boldsymbol{\lambda}/\|\boldsymbol{\lambda}\|_1) \big/ \log D = \Big( - \sum_i (\lambda_i/\|\boldsymbol{\lambda}\|_1) \log(\lambda_i/\|\boldsymbol{\lambda}\|_1) \Big) \Big/ \log D \qquad (11)$$

where $\boldsymbol{\lambda}$ denotes the eigenvalues of the covariance matrix of $\mathbf{X}_{c_1\|c_2}$, and $\mathbf{X}_{c_1\|c_2} \in \mathbb{R}^{|\mathcal{B}_{c_1\|c_2}| \times D}$ is formed by the $\phi_{L-1}(G_{c_1\|c_2})$, $\forall\, G_{c_1\|c_2} \in \mathcal{B}_{c_1\|c_2}$. The embedding of the last hidden layer $\phi_{L-1}$ is chosen because this measurement of complexity is only applicable to linearly separable boundaries. For each adjacent pair, the complexity measure is presented in Table 1. It is interesting to observe that the boundaries of the GNN trained on Enzymes exhibit the highest complexity among all the explained GNNs, while the boundaries of the GNN for Motif are the simplest. In general, the real-world datasets are anticipated to be more challenging and the tasks involving more classes can even make the decision-making rules more complex. Therefore, the observation that the GNN trained on Enzymes possesses the most intricate boundaries closely aligns with our expectations.

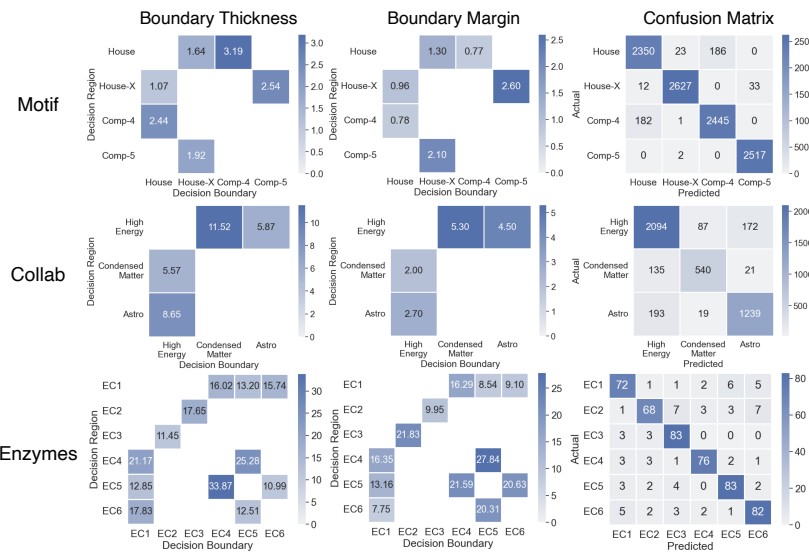

Figure 3: Three metrics for analyzing the decision boundaries of GNNs trained on three datasets.

## 6 CONCLUSION

In this paper, we take the first step toward explaining the GNNs through the lens of decision boundaries. We propose a model-level explainability method, GNNBoundary, which attempts to understand the decision boundaries of GNNs by generating and analyzing boundary graphs. Specifically, we first develop an algorithm to identify the pairs of classes that have adjacent decision regions. Given the adjacent class pairs, we generate faithful near-boundary graphs by optimizing a novel objective function tailored to satisfy the two desired properties we propose for boundary graph generation. The experimental results on both real-world and synthetic datasets have shown that we could consistently generate faithful near-boundary graphs. More importantly, our case studies illustrate that the boundary graphs generated by GNNBoundary can be utilized to measure the boundary thickness, boundary margin, and complexity of the boundary. In essence, through the analysis of boundary graphs generated by GNNBoundary, we can obtain profound insights into the robustness to perturbation, out-of-distribution robustness, and generalizability of GNNs. These insights would pave the way for establishing unwavering trust in decisions made by GNNs and encouraging increasing deployment of GNNs in real-world applications.

## 7 ACKNOWLEDGEMENT

The work was supported in part by the US Department of Energy SciDAC program DE-SC0021360, National Science Foundation Division of Information and Intelligent Systems IIS-1955764, and National Science Foundation Office of Advanced Cyberinfrastructure OAC-2112606.

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

# Appendix

## Table of Contents

## A  PROOF OF PROPOSITION 4.1

*Proof.* Let $\mathbf{x} = \phi_{L-1}(G)$ be the logits predicted by $f$. The cross-entropy loss function is defined as

$$\mathcal{L} = -\sum_{c \in [1,C]} (\mathbf{p}_{\text{target}})_c \log \sigma(\mathbf{x})_c \tag{12}$$

Then, we have

$$\mathcal{L} = -\sum_{c \in [1,C]} (\mathbf{p}_{\text{target}})_c \log \sigma(\mathbf{x})_c \tag{13}$$

$$= -\sum_{c \in [1,C]} (\mathbf{p}_{\text{target}})_c \cdot \log \frac{e^{x_c}}{\sum_k e^{x_k}} \tag{14}$$

$$= -\sum_{c \in [1,C]} (\mathbf{p}_{\text{target}})_c \cdot \big(x_c - \text{logsumexp}(\mathbf{x})\big) \tag{15}$$

$$= \text{logsumexp}(\mathbf{x}) - \mathbf{p}_{\text{target}} \cdot \mathbf{x} \tag{16}$$

$$\tag{17}$$

Take the gradient of the function $\mathcal{L}$ with respect to $\mathbf{x}$

$$\nabla_{\mathbf{x}}\mathcal{L} = \frac{\partial}{\partial \mathbf{x}} \text{logsumexp}(\mathbf{x}) - \mathbf{p}_{\text{target}} \cdot \mathbf{x} \tag{18}$$

$$= \text{Softmax}(\mathbf{x}) - \mathbf{p}_{\text{target}} \tag{19}$$

In the specific case of finding boundary graphs for $c_1$ and $c_2$, we have

$$\frac{\partial \mathcal{L}}{\partial x_c} = \begin{cases} \sigma(\mathbf{x})_c - 0.5, & c \in \{c_1, c_2\} \\ \sigma(\mathbf{x})_c, & c \notin \{c_1, c_2\} \end{cases} \tag{20}$$

Next, let's compute the partial derivatives of Equation 2,

$$\nabla_{\mathbf{x}}\mathcal{L}' = \frac{\partial}{\partial \mathbf{x}} - \sum_{b \in \{c_1,c_2\}} f(G)_b \cdot \big(0.5 - p^*(b)\big) + \sum_{b' \notin \{c_1,c_2\}} f(G)'_b \cdot p^*(b') \tag{21}$$

$$= \frac{\partial}{\partial \mathbf{x}} \sum_{b \in \{c_1,c_2\}} x_b \cdot \big(p^*(b) - 0.5\big) + \sum_{b' \notin \{c_1,c_2\}} x_{b'} \cdot p^*(b') \tag{22}$$

$$\tag{23}$$

Then we can get

$$\frac{\partial \mathcal{L}'}{\partial x_c} = \begin{cases} p^*(c) - 0.5, & c \in \{c_1, c_2\} \\ p^*(c), & c \notin \{c_1, c_2\} \end{cases} \tag{24}$$

$$= \begin{cases} \sigma(\mathbf{x})_c - 0.5, & c \in \{c_1, c_2\} \\ \sigma(\mathbf{x})_c, & c \notin \{c_1, c_2\} \end{cases} \tag{25}$$

$$\frac{\partial \mathcal{L}'}{\partial x_c} = \frac{\partial \mathcal{L}}{\partial x_c}, \forall c \tag{26}$$

Therefore, with gradient back-propagation, minimizing cross-entropy loss defined by Equation 1 is the same as minimizing Equation 2.

$\square$

## B    Experimental Details

The experiments were conducted on an Apple M1 Max processor. In terms of software configuration, the models were developed using Python 3.10. For auto-differentiation and numerical optimization tasks, PyTorch 2.0 was utilized. The Graph Convolutional Network (GCN) architecture was selected for both the Collab and Motif datasets and was constructed using the PyTorch Geometric 2.2 framework. Vector gather-scatter operations were managed using the PyTorch-Scatter library. Additionally, the generation and manipulation of graph data were facilitated through the NetworkX library. Regarding hyperparameter configurations, we set the temperature of the Concrete Distribution $\tau$ to 0.15. The sample size $K$ for every Monte Carlo sampling is set at 32. For the optimization procedure, we employed the Stochastic Gradient Descent (SGD) optimizer with a learning rate of 1. Pertaining to the regularization weights for $R_{L_1}$ and $R_{L_2}$, both are consistently set to 1 across all experimental runs.

## C    Comparison with Cross-Entropy Loss

Table 2: The quantitative comparison in terms of convergence speed between the proposed adaptive boundary loss and the cross-entropy loss. The success rate measures how many times the near-boundary graphs can be successfully generated within 500 iterations of optimization over 1000 runs. The average convergence iteration is the expected number of iterations required for convergence over all the successful runs out of 1000. The numbers in bold indicate the superior performance.

| Dataset | $c_1$ | $c_2$ | Success Rate | | Average Convergence Iteration | |
|---|---|---|---|---|---|---|
| | | | GNNBoundary | Cross-entropy Baseline | GNNBoundary | Cross-entropy Baseline |
| Motif | House | HouseX | **0.86** | 0.80 | **136.25** | 180.30 |
| | House | Comp4 | 1.00 | 1.00 | **20.44** | 24.69 |
| | HouseX | Comp5 | **0.76** | 0.55 | 178.95 | **158.76** |
| Collab | HE | CM | **1.00** | 0.95 | **46.62** | 50.80 |
| | HE | Astro | **1.00** | 0.98 | **15.70** | 17.07 |
| Enzymes | EC1 | EC4 | **0.63** | 0.47 | **177.79** | 224.60 |
| | EC1 | EC5 | **0.96** | 0.87 | 88.39 | **87.12** |
| | EC1 | EC6 | **0.75** | 0.25 | **138.75** | 180.00 |
| | EC2 | EC3 | **0.74** | 0.66 | **136.43** | 172.39 |
| | EC4 | EC5 | **0.57** | 0.43 | **167.71** | 251.79 |
| | EC5 | EC6 | **1.00** | 0.98 | **80.40** | 115.82 |

Even though there is no existing method for understanding the decision boundaries of GNNs, we still manage to compare our performance with some proper baseline approaches. In addition to the random baseline approach mentioned in Section 5.2 of the manuscript, we also evaluate the comparative performance of GNNBoundary with the proposed adaptive loss function against GNNBoundary

with the cross-entropy loss. Cross-entropy loss is a commonly used objective function for generating boundary cases of Deep Neural Networks or Convolutional Neural Networks Berk et al. (2022). However, based on Proposition 4.1 and Corollary 4.1, minimizing the cross-entropy loss has a limitation that the logit value of boundary classes will be discouraged in some scenarios, which hinders the convergence during the optimization. Thus, we propose an adaptive loss function (see Equation 3) to mitigate this issue such that optimizing the proposed adaptive loss function is less likely to be trapped in local minima without converging. Therefore, in this section, we conducted experiments to empirically compare the convergence speed of our loss function and cross-entropy loss function.

To evaluate the comparative performance, we assessed the convergence speed by measuring the success rate and average convergence rate. Namely, the success rate measures how many times the boundary graphs satisfying the near-boundary criterion in Equation 7 can be successfully generated within 500 iterations; the average convergence rate is the expected number of iterations required for convergence, if the near-boundary graphs are successfully generated within 500 iterations. We evaluated these two metrics by optimizing both loss functions for 1000 runs. As shown in Table 2, for all adjacent pairs in all three datasets, GNNBoundary obtains a significantly higher success rate within 500 iterations. Also, comparing the average convergence iteration, GNNBoundary can converge within a substantially smaller number of iterations on average, for almost all adjacent pairs across three datasets. In conclusion, this empirical result has demonstrated that the adaptive boundary loss function we proposed can effectively and consistently generate near-boundary graphs with faster convergence and reduced likelihood of being trapped in local minima.

# D  QUALITATIVE COMPARISON WITH GNNINTERPRETER

Given the fact that GNNBoundary is a model-level explanation method for GNNs, it would be meaningful if we could compare it with the existing model-level explanation methods. The existing model-level explainability method often focuses on extracting discriminative features for each class in the graph classification task. However, different from other model-level explanation methods, GNNBoundary focuses on understanding the decision boundaries of GNNs. Obviously, we have a totally different objective than the existing model-level explanation methods so the quantitative comparison might not make sense here. Therefore, in this section, we aim to qualitatively compare with an existing model-level explanation method called GNNIntrepreter (Wang & Shen, 2023). In Figure 4, we present the qualitative examples of boundary graphs generated by GNNBoundary, and the examples of explanation graphs generated by GNNInterpreter.

In general, qualitatively analyzing the explanation graphs for GNNs trained on real-world datasets would be very challenging, because the ground-truth class features for most real-world graph classification datasets are not known. Otherwise, we can directly use some greedy rule-based algorithms to classify the graphs without the need to train a GNN model. Therefore, for real-world datasets, humans cannot form a meaningful expectation for the generated explanation graphs since the ground truth features of each class are unknown. For this reason, our qualitative analysis would mainly focus on the synthetic dataset, Motif.

Through the qualitative analysis of the generated boundary graphs, we indeed observed several interesting patterns indicating that the boundary graphs share some commonalities with both boundary classes but do not belong to any of them. For example, the boundary graph between House-X and Comp-5 has much denser edge connectivity than the boundary graph between House-X and House, even though those two boundary graphs are both predicted as House-X with almost 0.5 probability. For another example, we observed that the boundary graph between House and Comp-4 contains only a single purple node while the boundary graph between House and House-X has 4 purple nodes, which may be caused by the fact that Comp-4 does not have a purple node but both House-X and House has a purple node. However, we also realized that the boundary graphs generated by GNNBoundary might not be fully consistent with the general expectations of humans regarding the boundary graphs to some extent. This actually indicates a potential discrepancy between the boundary graphs in the belief of humans and the boundary graphs in the belief of the models. In other words, the boundary graphs presented in Figure 4, in fact, reflect the difference in the decision-making rules between GNNs and humans.

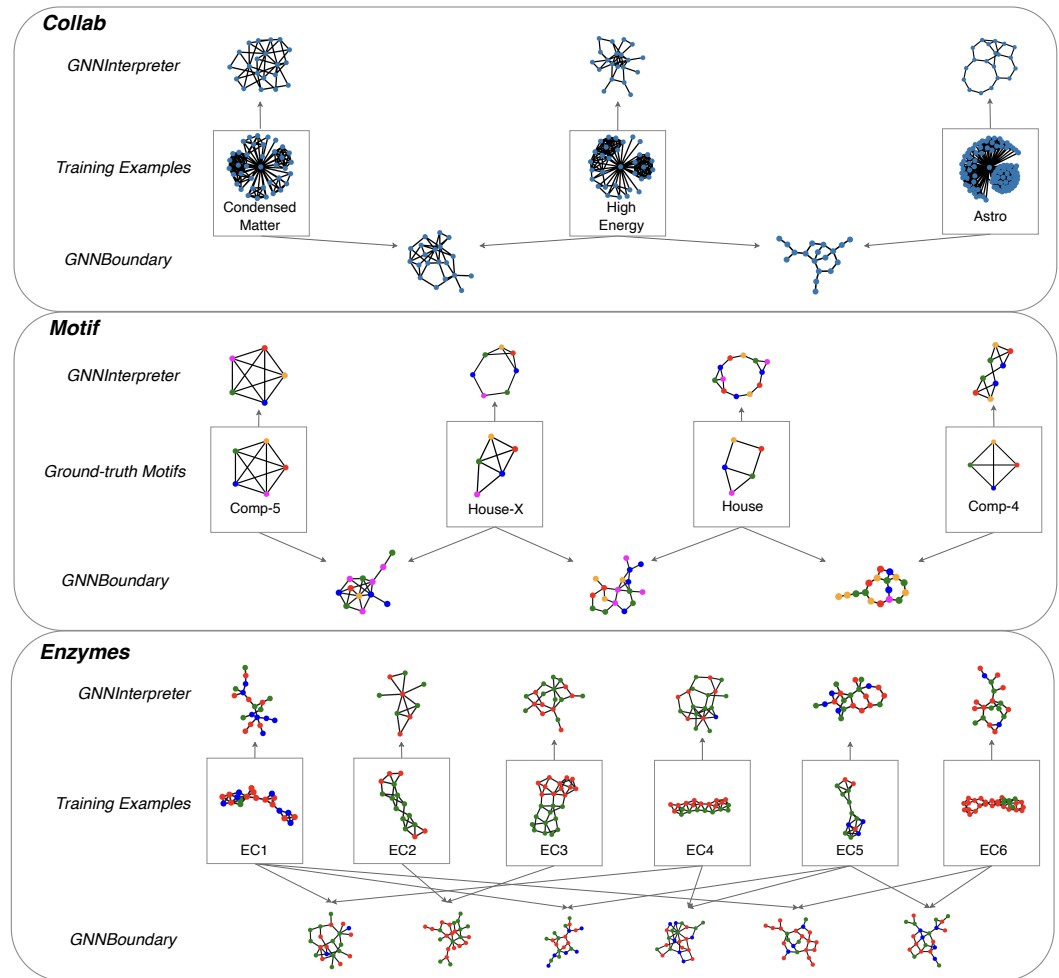

Figure 4: The qualitative comparison of explanation graphs between GNNBoundary and GNNINterpreter. In addition to the explanation graphs, we also present an example of a true graph for each class in the Collab and Enzymes dataset, and the ground-truth motif corresponding with each class in the Motif dataset.

Speaking of GNNInterpreter, since their explanation graph is generated to trigger a specific response (the class of interest) from the GNN as much as possible, their explanation graph is supposed to encapsulate all the discriminative features of the target class. To be specific, the explanation graphs generated by GNNInterpreter will be classified by the GNNs as the target class with a probability of 1; the explanation graphs generated by GNNBoundary will be classified by the GNNs as two boundary classes with equal probability. Therefore, we should set a completely different expectation for the explanation graphs generated by GNNInterpreter. For House-X, House, and Comp-4 in the Motif dataset, we can observe a clear discrepancy between the ground-truth motif and the explanation graphs generated by GNNInterpreter. In other words, the GNN trained on Motif will misclassify the explanation graphs generated by the GNNInterpreter since they actually do not contain the ground-truth motifs. This is similar to what we observed in the boundary graph generated by GNNBoundary, which indicates a potential flaw in the decision-making rules of GNNs.

