# OpenReview forum: "GNNBoundary: Towards Explaining Graph Neural Networks through the Lens of Decision Boundaries"
_ICLR.cc/2024/Conference — ICLR 2024 poster_

### Official Review · Reviewer_F52p · 2023-10-29

**Soundness:** 3 good
**Presentation:** 3 good
**Contribution:** 2 fair
**Rating:** 6
**Confidence:** 3

**Summary:**

This paper introduces GNNBoundary, a  methodology designed to generate boundary graphs for Graph Neural Networks (GNNs), aiming to provide a deeper understanding of their decision boundaries. The GNNBoundary first identify the adjacent classes and then generate boundary graphs with cross-entropy loss.  The authors conduct extensive experiments and case studies on both synthetic and real-world datasets to validate the effectiveness of GNNBoundary. The generated boundary graphs are utilized to analyze various aspects of decision boundaries, including boundary thickness, margin, and complexity.

**Strengths:**

1. The paper addresses an important and timely topic in the domain of GNNs, focusing on explainability and interpretability, which are crucial for the adoption of GNNs in real-world applications.
2. The paper is well-structured and easy to follow.
3. The paper provides a thorough evaluation of GNNBoundary, and the results seem to demonstrate the effectiveness of the approach.

**Weaknesses:**

1. In Section 3, the decision boundary is characterized by a scenario where a graph $G$ is equally probable to belong to classes $c_1$ and $c_2$. However, achieving this exact equality in probabilities is practically challenging. A more feasible approach might be to define the decision boundary within a specific interval or range of probabilities, ensuring a more robust and attainable characterization.

2. In Section 4.2, the authors state: "if traversal through decision region $R_{c3}$ is necessary, then $R_{c1}$ and $R_{c2}$ are not adjacent." However, the linkage between this statement and Algorithm 1 is not immediately clear. To enhance comprehension, it would be beneficial for the authors to provide a more explicit explanation or illustration of how this concept is integrated into or related to the steps and operations outlined in Algorithm 1.

3. There is no formal math definition of adjacency of decision region. The adjacency of decision region should be defined on the input space i.e., the graph sapce. However, the degree of adjacenct of class pair in algorithm 1 is only a quantitively approximation of the adjacency and obtained on the embedding space.

4. Although the methods use some techniques to generate discrete node feature and edges, the degree of graphs should be varying even in one dataset. How do you determin the degree of boundary graphs which seems can only be pre-determined.

5. The proposed GNNBoundary methods seems not be specific for graph fields. The author should compare it with other methods in more general fields such as  [Berk et al., 2022], discuss its applicability in more general fields. Also the author could concentrate more on graph since the name of method is GNNBoundary.


6. There are some other typos such as the subscript of $\phi$ in Line 4 of algorithm have different meaning.

**Questions:**

See weakness.

---

> ### Author Response · Authors · 2023-11-23
> **Response to Reviewer F52p [1/3]**
>
> We sincerely appreciate your valuable time and efforts in reviewing this paper. We thank the thoughtful feedback you provided, which significantly improved the quality of this paper. For the potential concerns you bring up, we would like to answer and address them here.
>
> > In Section 3, the decision boundary is characterized by a scenario where a graph $G$ is equally probable to belong to classes $c_1$ and $c_2$. However, achieving this exact equality in probabilities is practically challenging. A more feasible approach might be to define the decision boundary within a specific interval or range of probabilities, ensuring a more robust and attainable characterization.
>
> Thanks for bringing up this lack of clarity. We strongly agree that obtaining the boundary graphs exactly lying on the boundary is practically challenging. This is exactly what we observed when conducting the experiment of GNNBoundary. Therefore, in Section 4.4 of the original manuscript, we introduced a near-boundary criterion which defines a criterion to determine whether the boundary graphs fall into an interval [$\alpha$,$\beta$]. As demonstrated in Algorithm 2, the optimization procedure will terminate once the condition at line 6 is met. In other words, once the sampled graphs fall into the predefined interval [$\alpha$,$\beta$], the learned graph distribution will be returned at line 7 of Algorithm 2. We have to admit that, the wording we previously used might be misleading because there are several times we mentioned that "GNNBoundary generates graphs that lie on the decision boundary". **In the revision, we have changed them to "extremely close to the decision boundary".**
>
> > In Section 4.2, the authors state: "if traversal through decision region $R_{c3}$ is necessary, then $R_{c1}$ and $R_{c2}$ are not adjacent." However, the linkage between this statement and Algorithm 1 is not immediately clear. To enhance comprehension, it would be beneficial for the authors to provide a more explicit explanation or illustration of how this concept is integrated into or related to the steps and operations outlined in Algorithm 1.
>
> **In the revised manuscript, we rewrote the entire Section 4.1 for better comprehension as you suggested.** We sincerely appreciate your helpful suggestions.
>
> > There is no formal math definition of adjacency of decision region. The adjacency of decision region should be defined on the input space i.e., the graph sapce. However, the degree of adjacenct of class pair in algorithm 1 is only a quantitively approximation of the adjacency and obtained on the embedding space.
>
> **Following your suggestions, we introduced a formal math definition of adjacency of decision region in Section 4.1** Based on this formal math definition, we can show why the degree of adjacency measured in Algorithm 1 can serve as an approximation of adjacency of decision region.

---

> ### Author Response · Authors · 2023-11-23
> **Response to Reviewer F52p [2/3]**
>
> > Although the methods use some techniques to generate discrete node feature and edges, the degree of graphs should be varying even in one dataset. How do you determine the degree of boundary graphs which seems can only be pre-determined.
>
> Thanks for your insightful comments. It is true that the graphs in the same dataset will have different numbers of nodes and different edge connections. However, we carefully designed our approach such that the degree of boundary graphs does not need to be manually pre-specified for generating every different boundary graph. As mentioned in Section 4.3, we assume the boundary graph is a Gilbert random graph $G(n,\mathbf{\theta})$ where $n$ denotes the number of nodes and $\mathbf{\theta}$ denotes the probability that every possible edge independently occurs. Then, the boundary graph distribution is continuously relaxed and re-parameterized using the Gumbel-Softmax trick such that the parameter of the boundary graph distribution can be learned through gradient descent. Once learned, the expected boundary graph can be obtained by taking all the edges with the probability $\theta_{ij} > 0.5$. In other words, $n$ is the maximum number of nodes in the boundary graphs, which is specified as 20 for all datasets in our experimental study, but the actual number of nodes and edges in the expected boundary graph is determined by the learned $\mathbf{\theta}$. The qualitative examples of expected boundary graphs are presented in Figure 4 of Appendix D. As you can see, for the same dataset, the generated boundary graphs may have different numbers of nodes with different edge connections, without the need to manually pre-specify the number of nodes and number of edges.
>
> > There are some other typos such as the subscript of $\phi$ in Line 4 of algorithm have different meaning.
>
> Thanks for pointing this out. **The typos for subscript of $\phi$ have been corrected such that it follows a consistent meaning. Additionally, we carefully reviewed the entire manuscript and corrected the inappropriate wordings, typos, and inconsistencies in the notations, ensuring that the paper is grammatically correct, clear, and easy to read.**

---

> ### Author Response · Authors · 2023-11-23
> **Response to Reviewer F52p [3/3]**
>
> > The proposed GNNBoundary methods seems not be specific for graph fields. The author should compare it with other methods in more general fields such as [Berk et al., 2022], discuss its applicability in more general fields. The author could concentrate more on graph since the name of method is GNNBoundary.
>
> Thanks for your insightful comments. We fully understand why you believe GNNBoundary can be generalized to other fields because the loss function we proposed and the algorithm to identify adjacent class pairs might be a general approach for other fields such as image classification. However, we would like to clarify that, the end-to-end explanation pipeline of GNNBoundary is only applicable to explain GNNs. **To clearly illustrate our overall procedure, we added a figure (see Figure 1) in the revised manuscript to present the high-level overview of GNNBoundary.** In Figure 1, the boundary graph distribution we defined, the continuous relaxation of graphs, the optimization method via the Gumbel-Softmax trick, and the dynamic regularization scheduler we proposed, are only applicable to graph fields. Especially for the dynamic regularization scheduler we proposed in Section 4.4, it is only applicable to graphs because only graph data can have various numbers of nodes and edges. Without constraining the graph size, the optimizer tends to generate boundary graphs without succinct graph features.
>
> It is true that the adaptive boundary loss function in Equation 3 and the algorithm for identifying adjacent pairs in Algorithm 1 might be applicable to other types of data. We would be more than happy to see that the proposed loss function and Algorithm 1 can be beneficial to more general fields. However, adapting our adaptive boundary loss function and the adjacency identification algorithm to other fields might not be a trivial task because different types of data have their unique characteristics. Also, it would require extensive experimental study to empirically prove their effectiveness in other fields. In a word, since our main focus is understanding the decision boundaries of GNNs and the end-to-end pipeline of GNNBoundary is not applicable to other fields, it might not be appropriate to extensively discuss the applicability of our loss function and adjacency identification algorithm in the manuscript. But we do believe that their applicability in other fields is very worthy of studying, as a part of our future research. Thank you for pointing us toward this interesting research direction.
>
> **Even though GNNBoundary is not applicable to other fields, as you suggested, we still managed to conduct a fair comparison with the explanation method for image classification proposed by [Berk et al., 2022] (see Appendix C).** The method they proposed is called U-DeepDIG. Compare GNNBoundary with U-DeepDIG, the only thing worth comparing is the loss function of generating the boundary instances. All other parts are only relevant to their own specific fields. Specifically, U-DeepDIG generates boundary instances by minimizing the cross-entropy loss in Equation 1. In theory, based on Proposition 4.1 (proved in Appendix A) and Corollary 4.1, minimizing the cross-entropy loss has a clear limitation that the logit value of boundary classes will be discouraged in some scenarios, which hinders the convergence during the optimization. Thus, we propose an adaptive loss function in Equation 2 to mitigate this issue such that optimizing the proposed adaptive loss function is less likely to be trapped in local minima without converging. Therefore, theoretically speaking, our loss function can generate boundary cases more effectively than the loss function of U-DeepDIG. This is also further verified by the experimental results presented in Table 2. To evaluate the comparative performance, we assessed convergence speed by measuring the success rate and average convergence rate. Namely, the success rate measures how many times the boundary graphs satisfying the near-boundary criterion in Equation 7 can be successfully generated within 500 iterations; the average convergence rate is the expected number of iterations required for convergence, if the near-boundary graphs are successfully generated within 500 iterations. We evaluated these two metrics by optimizing both loss functions for 1000 runs. As shown in Table 2, for all adjacent pairs in all three datasets, GNNBoundary obtains a significantly higher success rate within 500 iterations. Also, comparing the average convergence iteration, GNNBoundary can converge within a substantially smaller number of iterations on average, for almost all adjacent pairs across three datasets. In conclusion, compared with the cross-entropy loss adopted by U-DeepDIG, this empirical result has demonstrated that the adaptive boundary loss function we proposed can effectively and consistently generate near-boundary graphs with faster convergence and less probability of being trapped in local minima.

---

### Official Review · Reviewer_oN5P · 2023-10-31

**Soundness:** 3 good
**Presentation:** 3 good
**Contribution:** 2 fair
**Rating:** 6
**Confidence:** 3

**Summary:**

In this paper, a novel model-level explanation method of GNNs called GNNBoundary is introduced, aimed at revealing the inner workings of GNNs by analyzing their decision boundaries. Unlike existing model-level methods, GNNBoundary approaches GNN explainability from a fresh perspective by identifying adjacent class pairs and generating graphs that lie on decision boundaries. Through this approach, the paper enables a deeper understanding of the decision-making process of GNNs, assessing their robustness, generalizability, and the potential risk of misclassification on unseen data. The method is evaluated on synthetic and real-world datasets, demonstrating its effectiveness in generating faithful boundary graphs and providing valuable insights into decision boundaries' complexity and model robustness.

**Strengths:**

* The paper introduces an innovative model-level explanation method, GNNBoundary, which focuses on revealing the decision boundaries of GNNs. This approach offers a fresh perspective on explaining GNNs and has the potential to provide valuable insights into their behavior.
* The paper includes a comprehensive evaluation of GNNBoundary on both synthetic and real-world datasets. It provides quantitative and qualitative evidence of the method's efficiency and effectiveness in generating faithful boundary graphs and offering insights into GNN behavior.

**Weaknesses:**

* The paper only gives the boundary graph but not further analysis of how to use the boundary graph to analysis GNNs.
* The paper doesn't include relational graphs that include the most well-knowned graph tasks, molecule tasks.

**Questions:**

1. The boundary graph you showed is not so explicit why it's at the boundary. How will it help with GNN explanation? The case study part seems only considering the dataset part not the model part.
2. For graph-level tasks, one may think about tasks of molecules. They have edge types in the graph and can GNNboundary be applied to molecules? And now the boundary graph seems not similar to each of the class sample. Will it be better in the molecule fields?

---

> ### Author Response · Authors · 2023-11-23
> **Response to Reviewer oN5P [1/4]**
>
> We sincerely appreciate your valuable time and efforts in reviewing this paper. We thank the thoughtful feedback you provided, which significantly improved the quality of this paper. For the potential concerns you bring up, we would like to answer and address them here.
>
> > The paper only gives the boundary graph but not further analysis of how to use the boundary graph to analyze GNNs.
>
> Thanks for bringing up this lack of clarity. In the case studies (Section 5.3), we have conducted experiments on utilizing the generated boundary graphs to measure the boundary thickness, boundary margin, and boundary complexity. More specifically, we computed these three metrics following the definitions in the existing works.
>
> **Boundary Margin.** Intuitively speaking, considering the distance of each training point to the decision boundary, the boundary margin is defined as the smallest distance to the decision boundary among all training points in the same class [1]. Therefore, the boundary margin could effectively measure the model's robustness to input perturbation. Since the closed-form mathematical expression of the decision boundary learned by GNNs is intractable, the common practice is utilizing the boundary cases as an approximation of the decision boundary. Therefore, following [2], we computed the boundary margin given the boundary graphs generated by GNNBoundary as in Equation 9.
>
> **Boundary Thickness.** As defined in [2], the boundary thickness of a classifier measures the expected distance to travel along line segments between different classes across a decision boundary. Therefore, a graph classifier with a thicker boundary is more generalizable to unseen data. Again, we utilized the boundary graphs generated by GNNBoundary to approximate the decision boundary and compute the boundary thickness as in Equation 10.
>
> **Boundary Complexity.** As proposed in [3], it attempts to measure the complexity of the decision boundary using the entropy of eigenvalues of boundary cases. Its high-level idea is that if the boundary cases span over fewer dimensions, then the classifier is less likely to overfit. We computed the complexity based on the boundary graphs generated by GNNBoundary as in Equation 11.
>
> In conclusion, inspired by the existing works [1-3], we utilized the boundary graphs generated by GNNBoundary, which served as an approximation to the decision boundary learned by GNNs, to measure the boundary thickness, boundary margin, and boundary complexity. Thus, these three metrics can help us to assess the GNNs' robustness to input perturbation and generalizability to unseen data. We have to admit that, we should have made the use cases of generated boundary graphs to be more clear. **Therefore, We have revised the fourth paragraph of the introduction (Section 1) and the case study section (Section 5.3).** We hope that the potential usage of generated boundary graphs for analyzing the decision boundaries of the explained GNNs has been clarified after the revision.

---

> ### Author Response · Authors · 2023-11-23
> **Response to Reviewer oN5P [2/4]**
>
> > The boundary graph you showed is not so explicit why it's at the boundary. How will it help with GNN explanation?
>
> Thank you for bringing up this interesting discussion. We totally understand that the boundary graphs presented in Figure 2 of the original manuscript may not be fully consistent with your expectations. According to our empirical analysis, we indeed observed several interesting patterns indicating that the boundary graphs share some commonalities with both boundary classes but do not belong to any of them. For example, the boundary graph between House-X and Comp-5 has much denser edge connectivity than the boundary graph between House-X and House, even though those two boundary graphs are both predicted as House-X with almost 0.5 probability. For another example, we observed that the boundary graph between House and Comp-4 contains only a single purple node while the boundary graph between House and House-X has 4 purple nodes, which may be caused by the fact that Comp-4 doesn't have a purple node but both House-X and House has a purple node.
>
> However, we also acknowledge that the boundary graphs generated by GNNBoundary might not be fully consistent with humans' general expectations regarding the boundary graphs to some extent. This actually indicates a potential discrepancy between the boundary graphs in human's belief and the boundary graphs in the model's belief. In other words, the boundary graphs presented in Figure 2 of the original manuscript, in fact, reflect the difference in the decision-making rules between GNNs and humans. **In order to further assess the inconsistency in boundary graphs between human's belief and model's belief, we added the comparison with a simple baseline method (see Table 1) that generates boundary graphs more consistent with human's general belief.** To be specific, we randomly draw one training graph from class $c_1$ and one training graph from class $c_2$, and connect these two graphs with a randomly sampled edge. We designed this baseline method with a hypothesis that the boundary graphs with respect to the general human's belief should contain the discriminative features of both classes. The quantitative results of this baseline method in Table 1 have shown that the average prediction probability of both class $c_1$ and $c_2$ computed over the boundary graphs generated by this baseline is far away from 0.5. This means that the model doesn't consider the graphs generated by this baseline as the boundary graphs from the model's perspective. Thus, the inconsistency in boundary graphs between human's belief and model's belief indeed exists. This inconsistency is an important factor that strongly motivated us to take the first step toward understanding the decision boundaries of GNNs. In a word, since the boundary graphs generated by GNNBoundary represent the model's belief instead of the human's belief, we shouldn't expect the generated boundary graphs to be exactly consistent with boundary graphs in terms of human expectation.
>
> Speaking of how the boundary graphs can help to qualitatively explain the GNNs, this is not a trivial task and even deserves writing another paper.  To understand how the boundary graphs generated by GNNBoundary are topologically different from human expectation and what are the topological relationships between boundary graphs and training graphs, a method specifically designed for analyzing and summarizing the topological structures of boundary graphs is required. Without this method, the qualitative examples of boundary graphs might not convey clear and useful information about the decision boundaries. Thanks for asking this inspiring question which lights up an interesting direction for our future research. For this paper, since our main focus is generating boundary graphs that can be utilized to quantitatively analyze the decision boundaries (e.g., boundary complexity, boundary thickness, and boundary margin), **we decided to move the qualitative results from the manuscript into Appendix D, added more analysis about the commonalities between boundary graphs and true graphs, and carefully discussed the potential inconsistency it might indicate.**

---

> ### Author Response · Authors · 2023-11-23
> **Response to Reviewer oN5P [3/4]**
>
> > The case study part seems only considering the dataset part not the model part.
>
> Thanks for bringing this to our attention. **To clearly illustrate our overall procedure, we added a figure (see Figure 1) in the revised manuscript to present the high-level overview of GNNBoundary.** As shown in Figure 1, the only part of case studies that is relevant to the dataset is that the explained GNNs are trained on those datasets. Once the GNNs were trained, we didn't involve the dataset at all while conducting the case studies. Specifically, given the trained GNNs, we employ GNNBoundary to generate the boundary graphs that obtain the same probability of class $c_1$ and $c_2$ as predicted by the GNNs. Then, we generated graphs in the decision regions of all classes by GNNInterpreter. Given the boundary graphs generated by GNNBoundary and the graphs in the decision regions generated by GNNInterpreter, we computed the boundary thickness, boundary margin, and boundary complexity. Because the boundary graphs generated by GNNBoundary can be regarded as an approximation of decision boundaries of GNNs, these three computed metrics, by definitions, can be used to assess the GNNs' robustness to input perturbation, GNNs' generalizability, and GNNs' potential risk of misclassification on unseen data.
>
> > The paper doesn't include relational graphs that include the most well-knowned graph tasks, molecule tasks.
>
> Thanks for your helpful suggestions. **In Section 5 of the revised manuscript, we have conducted an additional experiment on a real-world molecule dataset called Enzyme.** Enzymes contains 6 classes and 3 types of nodes, and each class corresponds to one type of enzyme. Note that, Enzymes dataset does not have different edge types. From Table 1, we can see that GNNBoundary can consistently generate faithful near-boundary graphs for all adjacent pairs identified in Enzymes. In addition, based on the complexity we computed over the generated boundary graphs, Enzymes possess the most complicated decision boundaries among all three datasets. The potential reason is that the complexity of boundaries tends to grow while the number of classes increases. In conclusion, this additional experiment conducted on Enzymes not only showcases our effectiveness in explaining a well-known real-world molecule classification task, but also demonstrates our capability in explaining the graph classifier with a large number of classes.
>
>
> > For graph-level tasks, one may think about tasks of molecules. They have edge types in the graph and can GNNboundary be applied to molecules? And now the boundary graph seems not similar to each of the class sample. Will it be better in the molecule fields?
>
> Thanks for your thoughtful comments. After scanning through this exhaustive list of real-world graph classification datasets (https://ls11-www.cs.tu-dortmund.de/staff/morris/graphkerneldatasets), we observed that all the molecule datasets with edge types are binary classification tasks. However, since our main focus is to understand the decision boundaries of GNNs, a graph classifier with only two classes is trivial to explain. Therefore, considering the constructive feedback from you and Reviewer hgmZ, we carefully chose Enzymes as the additional experiment because this is a
> molecule dataset with the greatest number of classes among all real-world graph classification datasets.
>
> Regarding your concerns about interpreting the qualitative examples of generated boundary graphs,  the situation is actually not better for Enzymes as shown in Figure 4 of Appendix D. As mentioned in our earlier response, we shouldn't expect the generated boundary graphs to be fully consistent with human expectations, because there is an inconsistency in the decision-making rules between humans and GNNs. In addition, qualitatively analyzing the boundary graphs for GNNs trained on real-world datasets would be very challenging, because the ground-truth class features for most real-world graph classification datasets are not known. Otherwise, we can directly use some greedy rule-based algorithms to classify the graphs without the need to train a GNN model. Therefore, for real-world datasets, humans cannot form a meaningful expectation for the generated boundary graphs since the ground truth features of each class are unknown. That is the reason why we conducted our experiments on both real-world
> datasets and synthetic datasets. Experiments on synthetic datasets would be easier to classify but have better interpretability given that the ground-truth features are pre-determined by us, while experiments on real-world datasets can demonstrate our effectiveness in explaining GNNs that tackle complex real-world problems in various different fields (e.g., chemistry and social science).

---

> ### Author Response · Authors · 2023-11-23
> **Response to Reviewer oN5P [4/4]**
>
> ## Reference
>
> [1] Elsayed, Gamaleldin, et al. "Large margin deep networks for classification." Advances in neural information processing systems 31 (2018).
>
> [2] Yang, Yaoqing, et al. "Boundary thickness and robustness in learning models." Advances in Neural Information Processing Systems 33 (2020): 6223-6234.
>
> [3] Guan, Shuyue, and Murray Loew. "Analysis of generalizability of deep neural networks based on the complexity of decision boundary." 2020 19th IEEE International Conference on Machine Learning and Applications (ICMLA). IEEE, 2020.

---

> > ### Comment · Reviewer_oN5P · 2023-11-23
> > **Response the the Authors**
> >
> > I appreciate the authors' reponses and they address my concerns adequately. I will raise my score to 6

---

### Official Review · Reviewer_9PKK · 2023-10-31

**Soundness:** 3 good
**Presentation:** 3 good
**Contribution:** 3 good
**Rating:** 8
**Confidence:** 3

**Summary:**

The work studies explainability for GNNs by introducing  the notion of boundaries between predicted classes. The core of the contribution is to propose a method to find the boundaries, both which class appears to be adjacent to which, and to find examples of graphs on thes boundaries. Then the authors study some metrics computed on these boundaries (margin, thickness, complexity) that contribute to model-level explainability for GNNs.

**Strengths:**

- The idea of explainability through the decision boundaries is simple, very elegant and relevant and, as far as I know, novel for GNNs.

- The work is well conducted (one understands well how algorithm 1 and then 2 are built), justified from the existing literature, and globally well written.

- The derivation of the objective function, eq. (5), is nicely done, showing the limitation one would have by using directly the cross-entropy loss.

- The numerical experiments are adequate.

**Weaknesses:**

- The qualitative interpretation of the results, that are especially elaborated from the study of 2 datasets and Figure 2 that shows what classes there are and what are the boundaries, is difficult to fully agree with.


- There is no real comparison of the elements of explainability for GNNs obtained by the proposed method to what the existing, yet different, explainability methods provide.
The work proposes, in 5.3, only a study of the metrics of the present work in some use case.

**Questions:**

- I don't see anything intuitive in the case of Collab graphs. Is this dataset really a good one to illustrate the method ? We don't have ground truth for the elements of explainability in this case, do we ?

- For Motif dataset, we have more intuition. Yet, I don't see why it's normal that they is no boundary between Comp-4 and House-X. There is only 1 added node and I wouldn't have said that the GNN rely that much to the number of nodes. Also, is it possible to understand the shape of the graphs on the found boundaries ?

---

> ### Author Response · Authors · 2023-11-23
> **Response to Reviewer 9PKK [1/3]**
>
> We sincerely appreciate your valuable time and efforts in reviewing this paper. We thank the thoughtful feedback you provided, which significantly improved the quality of this paper. For the potential concerns you bring up, we would like to answer and address them here.
>
> > The qualitative interpretation of the results, that are especially elaborated from the study of 2 datasets and Figure 2 that shows what classes there are and what are the boundaries, is difficult to fully agree with.
>
> Thanks for your thoughtful comments. In the original manuscript, we said that "the qualitative examples of boundary graphs (shown in Figure 2) are consistent with our expectation that they share some commonality with both boundary classes but do not belong to any of them". We totally understand that the presented qualitative examples may not be fully consistent with your expectations and the commonalities with both boundary classes may not be too clear. Regarding the commonalities we mentioned in the paper, we indeed observed several interesting patterns indicating that the boundary graphs share some commonalities with both boundary classes. For example, the boundary graph between House-X and Comp-5 has much denser edge connectivity than the boundary graph between House-X and House, even though those two boundary graphs are both predicted as House-X with almost 0.5 probability. For another example, we observed that the boundary graph between House and Comp-4 contains only a single purple node while the boundary graph between House and House-X has 4 purple nodes, which may be caused by the fact that Comp-4 doesn't have a purple node but both House-X and House has a purple node.
>
> However, we have to admit that the boundary graphs generated by GNNBoundary (shown in Figure 2 of the original manuscript) might not be fully consistent with humans' general expectations regarding the boundary graphs to some extent. This actually indicates a potential discrepancy between the boundary graphs in human's belief and the boundary graphs in the model's belief. In other words, the boundary graphs presented in Figure 2 of the original manuscript, in fact, reflect the difference in the decision-making rules between GNNs and humans. **In order to further assess the inconsistency in boundary graphs between human's belief and model's belief, we added the comparison with a simple baseline method (see Table 1) that generates boundary graphs more consistent with human's general belief.** To be specific, we randomly draw one training graph from class $c_1$ and one training graph from class $c_2$, and connect these two graphs with a randomly sampled edge. We designed this baseline method with a hypothesis that the boundary graphs with respect to the general human's belief should simultaneously contain the discriminative features of both classes. The quantitative results of this baseline method in Table 1 have shown that the average prediction probability of both class $c_1$ and $c_2$ computed over the boundary graphs generated by this baseline is far away from 0.5. This means that the model doesn't consider the graphs generated by this baseline as the boundary graphs from the model's perspective. Thus, the inconsistency in boundary graphs between human's belief and model's belief indeed exists. This inconsistency is an important factor that strongly motivated us to take the first step toward understanding the decision boundaries of GNNs. In a word, since the boundary graphs generated by GNNBoundary represent the model's belief instead of the human's belief, we shouldn't expect the generated boundary graphs to be exactly consistent with boundary graphs in terms of human expectation.
>
> Lastly, to understand how the boundary graphs generated by GNNBoundary are topologically different from human expectation and what are the topological relationships between boundary graphs and training graphs, a method specifically designed for analyzing and summarizing the topological structures of boundary graphs is required. Without this method, the qualitative examples of boundary graphs might not convey clear and useful information about the decision boundaries. However, designing this method is not a trivial task and deserves writing another paper, which lights up an interesting direction for our future research. For this paper, since our main focus is generating boundary graphs that can be utilized to quantitatively analyze the decision boundaries (e.g., boundary complexity, boundary thickness, and boundary margin), **we decided to move the qualitative results from the manuscript into Appendix D, added more analysis about the commonalities between boundary graphs and true graphs, and carefully discussed the potential inconsistency it might indicate.**

---

> ### Author Response · Authors · 2023-11-23
> **Response to Reviewer 9PKK [2/3]**
>
> > There is no real comparison of the elements of explainability for GNNs obtained by the proposed method to what the existing, yet different, explainability methods provide. The work proposes, in 5.3, only a study of the metrics of the present work in some use case.
>
> Thanks for your constructive suggestions. We strongly agree that comparing with existing but different methods would significantly improve the comprehensiveness of this paper. Unfortunately, since there is no existing work trying to understand the decision boundaries of GNNs, we don't have a proper existing method to compare with. However, we still tried our best to compare our method against some baseline approaches and the existing method in a proper way.
>
> First of all, we created two baseline approaches and compared against them, with the purpose of assessing the effectiveness of GNNBoundary. The first baseline approach, mentioned in our earlier response, can serve as a naive and random baseline that generates boundary graphs following human expectations. **The another baseline we created is replacing the proposed loss function in GNNBoundary with the cross-entropy loss (see Appendix C).** This baseline approach is meaningful because the cross-entropy loss is a commonly used objective function for generating boundary cases of Deep Neural Networks or Convolutional Neural Networks. In theory, we have proved in Section 4.2 that the proposed adaptive boundary loss is less likely to be trapped in local minima without converging, compared with the commonly used cross-entropy loss for generating boundary cases. Our empirical results presented in Table 2 are fully consistent with our theoretical proof. To evaluate the comparative performance, we assessed the convergence speed by measuring the success rate and average convergence rate. Namely, the success rate measures how many times the boundary graphs satisfying the near-boundary criterion in Equation 7 can be successfully generated within 500 iterations; the average convergence rate is the expected number of iterations required for convergence, if the near-boundary graphs are successfully generated within 500 iterations. We evaluated these two metrics by optimizing both loss functions for 1000 runs. As shown in Table 2, compared with the cross-entropy baseline, GNNBoundary obtains a significantly higher success rate within 500 iterations, while achieving a substantially smaller number of iterations on average. In conclusion, this empirical result has demonstrated that the adaptive boundary loss function proposed in this paper can effectively and consistently generate near-boundary graphs with faster convergence and less probability of being trapped in local minima.
>
> Secondly, even though there is no existing work on understanding the decision boundaries of GNNs, we still manage to compare with the existing explainability methods of GNNs. Since GNNBoundary is a model-level explainability method, it would be more appropriate if we could compare against with an existing model-level explainability method. The existing model-level explainability method often focuses on extracting discriminative features for each class in the graph classification task. Obviously, they have a totally different objective than GNNBoundary so the quantitative comparison might not make sense here. **Therefore, we added a qualitative comparison between the explanation graphs generated by GNNInterpreter (i.e., the current SOTA method of extracting the discriminative features for each class) and GNNBoundary in Appendix D.** To be specific, the explanation graphs generated by GNNInterpreter will be classified by the GNNs as the target class with a probability of 1; the explanation graphs generated by GNNBoundary will be classified by the GNNs as two boundary classes with equal probability. From Figure 4 in Appendix D, we can observe a clear discrepancy between the ground-truth motif and the explanation graphs generated by GNNInterpreter, for House-X, House, and Comp-4 in the Motif dataset. In other words, the GNN trained on Motif will misclassify the explanation graphs generated by the GNNInterpreter since they actually do not contain the ground-truth motifs. This is similar to what we observed in the boundary graph generated by GNNBoundary, which indicates a potential flaw in the decision-making rules of the GNN trained on Motif.

---

> ### Author Response · Authors · 2023-11-23
> **Response to Reviewer 9PKK [3/3]**
>
> > I don't see anything intuitive in the case of Collab graphs. Is this dataset really a good one to illustrate the method? We don't have ground truth for the elements of explainability in this case, do we?
>
> Thanks for bringing up this interesting discussion. It is true that the ground-truth features for each class in Collab graphs are not known. In fact, the ground-truth features for most real-world graph classification datasets are not known. Otherwise, we can directly use some greedy rule-based algorithms to classify the graphs without the need to train a GNN model. That is the reason why we conducted our experiments on both real-world datasets and synthetic datasets. Experiments on synthetic datasets would be easier to classify but have better interpretability given that the ground-truth features are pre-determined by us, while experiments on real-world datasets can demonstrate our effectiveness in explaining GNNs that tackle complex problems in real-world settings. Because training GNNs is actually not needed for synthetic datasets with known ground-truth features, the main focus of this paper is explaining GNNs on real-world datasets without known ground-truth features. **Therefore, in Section 5 of the revised manuscript, we conducted additional experiments on a real-world dataset called Enzyme, as suggested by reviewer hgmZ and reviewer oN5P.** Enzyme dataset is a molecule dataset containing 6 classes and 3 types of nodes, which can show our capability in explaining GNNs with a larger number of classes in the molecule field.
>
>
> > For Motif dataset, we have more intuition. Yet, I don't see why it's normal that they is no boundary between Comp-4 and House-X. There is only 1 added node and I wouldn't have said that the GNN rely that much to the number of nodes. Also, is it possible to understand the shape of the graphs on the found boundaries?
>
> We completely understand why you would expect Comp-4 and House-X to be adjacent. We actually had similar questions once we obtained these results. At that time, we carefully examined the 3D scatter plot of latent graph embedding output by the explained GNN after PCA projection. We observed that the latent graph embeddings of Comp-4 and House-X indeed are not adjacent. ** For your reference, we present this 3D scatter plot in this anonymous link (https://chart-studio.plotly.com/~gnnboundary/1).**  This interesting finding further reinforces our belief that understanding the decision boundaries of GNNs is an important task that is worth putting in more research efforts. Without understanding the decision boundaries, the GNNs' decision-making rules cannot be fully disclosed.
>
> In terms of analyzing the shape of the boundary graphs, it should involve profound topology analysis over the boundary graphs, and also investigate the topological similarity between boundary graphs and training graphs. This is a very crucial research direction because it would facilitate a deeper understanding of decision boundaries through a perspective different from GNNBoundary's. Thank you so much for pointing us to this exciting direction for our future research.
>
>
> ## Reference
>
> [1] J. Berk, M. Jaszewski, C. -A. Deledalle and S. Parameswaran, "U-Deepdig: Scalable Deep Decision Boundary Instance Generation," 2022 IEEE International Conference on Image Processing (ICIP), Bordeaux, France, 2022, pp. 2961-2965, doi: 10.1109/ICIP46576.2022.9897528.

---

### Official Review · Reviewer_hgmZ · 2023-11-01

**Soundness:** 2 fair
**Presentation:** 3 good
**Contribution:** 3 good
**Rating:** 6
**Confidence:** 3

**Summary:**

In this paper, the authors propose to investigate the explanation problem on graph classification tasks. Specifically, the authors propose to consider an innovative perspective from the decision boundaries of classification GNNs on graph classification. The authors, particularly, aim to generate graphs that exactly lie on the boundary such that practitioners can leverage such generated graphs for explanations to understand the behavior of the trained GNNs. The authors further conduct experiments on both synthetic and real-world datasets to demonstrate the effectiveness of their work.

**Strengths:**

1. The authors explore the important problem of explanations on graph classification tasks, which are crucial for a variety of real-world scenarios.

2. Originating from the innovative perspective of decision boundaries of GNNs between any pair of classes, the authors propose to generate graphs that exactly lie on the boundary based on the minimization of cross-entropy loss.

3. The experiments demonstrate the effectiveness of the proposed framework in explaining the classification GNNs.

**Weaknesses:**

1. Throughout the introduction, the authors do not provide sufficient references to support the claims proposed. For example, the first three sentences in the first paragraph do not involve any references, which will make the statement less reliable. Similar issues also exist in the third and fourth paragraphs.


2. The authors do not provide a thorough figure to illustrate the overall procedure of the proposed method. As a result, even though the core idea is straightforward, the process is somewhat unclear to readers.

3. The experiments are only conducted on two datasets, one being synthetic and another being real-world. The most important problem is that two datasets are not enough. Furthermore, these datasets are relatively small, and more importantly, the number of classes is extremely small (3 for Motif and 4 for COLLAB). We know that some graph classification datasets have only a small number of classes, however, as an explanation method, the authors should further explore the possibility of the method on larger datasets with a larger number of classes. This is also because the computation between different pairs of classes will significantly increase when the number of classes is larger.

4. As a novel method, we understand that the authors cannot find enough baselines for comparison. However, it is still possible to create several baselines for comparison. Otherwise, the comparison cannot be considered fair.

**Questions:**

Have the authors considered using more datasets, involving more graphs and classes? We are curious about the performance of the framework on such datasets. Our concern is that experiments on two datasets with 3 and 4 classes, respectively, are not convincing enough.

---

> ### Author Response · Authors · 2023-11-23
> **Response to Reivewer hgmZ [1/3]**
>
> We sincerely appreciate your valuable time and efforts in reviewing this paper. We thank the thoughtful feedback you provided, which significantly improved the quality of this paper. For the potential concerns you bring up, we would like to answer and address them here.
>
> >As a novel method, we understand that the authors cannot find enough baselines for comparison. However, it is still possible to create several baselines for comparison. Otherwise, the comparison cannot be considered fair.
>
> Thanks for your helpful suggestions. It is true that there is no existing method for explaining the decision boundaries of GNNs. However, we completely agree that creating several other baselines for comparison is necessary and helpful to improve the comprehensiveness of our evaluation study. **Therefore, in Section 5.2, we added a comparison regarding the faithfulness of boundary graphs against a random baseline approach; in Appendix C, we added a comparison regarding the optimization efficiency against a baseline approach with cross-entropy loss.**
>
> For the first baseline approach, it generates boundary graphs by connecting a randomly sampled pair of graphs, $G_{c_1} \in R_{c_1}$, $G_{c_2} \in R_{c_2}$ , with a random edge. It follows a general idea that the boundary graphs should contain the discriminative features of both $c_1$ and $c_2$. From Table 1, it shows that the boundary graphs generated by this baseline approach obtain an average prediction probability far away from 0.5 for all three datasets. However, the boundary graphs generated by GNNBoundary are extremely close to the decision boundaries of GNNs trained on all three datasets, because their predicted probabilities are all centered around 0.5. We can conclude that, GNNBoundary can generate much more faithful near-boundary graphs, compared with this baseline approach.
>
> Secondly, we created a baseline approach by replacing the proposed loss function in GNNBoundary with the cross-entropy loss. This baseline approach is meaningful because the cross-entropy loss is a commonly used objective function for generating boundary cases of Deep Neural Networks or Convolutional Neural Networks. In theory, we have proved in Section 4.2 that the proposed adaptive boundary loss is less likely to be trapped in local minima without converging, compared with the commonly used cross-entropy loss for generating boundary cases. Our empirical results presented in Table 2 are fully consistent with our theoretical proof. To evaluate the comparative performance, we assessed the convergence speed by measuring the success rate and average convergence rate. Namely, the success rate measures how many times the boundary graphs satisfying the near-boundary criterion in Equation 7 can be successfully generated within 500 iterations; the average convergence rate is the expected number of iterations required for convergence, if the near-boundary graphs are successfully generated within 500 iterations. We evaluated these two metrics by optimizing both loss functions for 1000 runs. As shown in Table 2, compared with the cross-entropy baseline, GNNBoundary obtains a significantly higher success rate within 500 iterations, while achieving a substantially smaller number of iterations on average. In conclusion, this empirical result has demonstrated that the adaptive boundary loss function proposed in this paper can effectively and consistently generate near-boundary graphs with faster convergence and less probability of being trapped in local minima.

---

> ### Author Response · Authors · 2023-11-23
> **Response to Reivewer hgmZ [2/3]**
>
> > The experiments are only conducted on two datasets, one being synthetic and another being real-world. The most important problem is that two datasets are not enough. Furthermore, these datasets are relatively small, and more importantly, the number of classes is extremely small (3 for Motif and 4 for COLLAB). We know that some graph classification datasets have only a small number of classes, however, as an explanation method, the authors should further explore the possibility of the method on larger datasets with a larger number of classes. This is also because the computation between different pairs of classes will significantly increase when the number of classes is larger.
>
> Thanks for your constructive comments. We strongly agree that explaining the decision boundaries of all possible pairs of classes is computationally expensive, especially for datasets with a large number of classes. However, GNNBoundary can be easily scaled to datasets with a large number of classes for two reasons.
>
> First, the number of class pairs of interest can be largely reduced by identifying the adjacent class pairs via Algorithm 1, since the decision boundaries only exist between the class pairs that have adjacent decision regions. To be more clear, we propose Algorithm 1 to measure the degree of adjacency for every possible pair of classes, such that the adjacent class pairs can be properly identified given the threshold value of the degree of adjacency. If the threshold value is high, then the number of adjacent class pairs of interest can be largely reduced. Speaking of its computational complexity, the complexity of Algorithm 1 is O($C^2$) where $C$ is the number of classes. Given this exhaustive list of real-world graph classification datasets (https://ls11-www.cs.tu-dortmund.de/staff/morris/graphkerneldatasets), there are only 10 percent of datasets have more than 10 classes. Therefore, given an algorithm with O($C^2$), our computational cost of identifying adjacent class pairs should not be expensive.
>
> Secondly, our algorithm (Algorithm 2) to generate the boundary graphs is more computationally efficient compared with other existing model-level explanation methods (i.e., XGNN [1] and D4Explainer [2]). For each adjacent class pair identified in Algorithm 1, we generate the boundary graphs via Algorithm 2 with the complexity of O($N^2$). However, the complexity of the reinforcement learning approach in XGNN [1] is O($M^3$) where $M$ is the number of edges; the complexity of the deep generative network in D4Explainer [2] is O($N^{2K}$) where $K$ is the number of edges to be modified and $N$ is the number of nodes. Even though their objective is not to generate boundary graphs, our generated boundary graphs can also be regarded as a form of model-level explanation graphs for interpreting GNNs. In a word, given the algorithm (Algorithm 1) to identify adjacent pairs and the efficient boundary graph generation method (Algorithm 2), GNNBoundary can be easily scaled to datasets with a large number of classes.
>
> Nevertheless, we totally agree that two datasets might not be sufficient to comprehensively demonstrate our effectiveness in explaining GNNs in various different fields. As suggested by Reviewer oN5P, the molecule task is the most well-known graph classification task, so conducting an additional experiment on molecule datasets would be a great plus to the comprehensiveness of our experimental study. **Therefore, considering the helpful suggestions from you and reviewer oN5P, we conducted an additional experiment on a real-world dataset called Enzyme in Section 5.** Enzymes contains 6 classes and 3 types of nodes. Each class corresponds to one type of enzyme. We carefully chose this dataset because Enzymes is the molecule dataset with the greatest number of classes among all real-world graph classification datasets (https://ls11-www.cs.tu-dortmund.de/staff/morris/graphkerneldatasets). From Table 1, we can see that GNNBoundary can consistently generate faithful near-boundary graphs for all adjacent pairs identified in Enzymes. In addition, based on the complexity we computed over the generated boundary graphs, Enzymes possess the most complicated decision boundaries among all three datasets. Its potential reason is that the complexity of boundaries tends to grow when the number of classes increases.

---

> ### Author Response · Authors · 2023-11-23
> **Response to Reivewer hgmZ [3/3]**
>
> > Throughout the introduction, the authors do not provide sufficient references to support the claims proposed. For example, the first three sentences in the first paragraph do not involve any references, which will make the statement less reliable. Similar issues also exist in the third and fourth paragraphs.
>
> Thanks for your kind suggestions. We strongly agree that we have missed some important references in our paper, which may potentially weaken our claims.  **Therefore, we added three references in the first paragraph of the introduction and two references in the third paragraph of the introduction.** For the fourth paragraph, its main focus is summarizing our method and contribution, without making any general claim. After thorough consideration, we believe it would be better to leave it as it is. **Additionally, we carefully scanned through the entire manuscript and added more references in Section 3 and Section 4, in order to better support our claims.**
>
> > The authors do not provide a thorough figure to illustrate the overall procedure of the proposed method. As a result, even though the core idea is straightforward, the process is somewhat unclear to readers.
>
> **Thanks to your suggestions, we designed a figure to illustrate our overall procedure (see Figure 1 in the revised manuscript).** We sincerely appreciate your helpful feedback for improving the readability of this paper.
>
> ## Reference
>
> [1] Yuan, Hao, et al. "Xgnn: Towards model-level explanations of graph neural networks." Proceedings of the 26th ACM SIGKDD International Conference on Knowledge Discovery & Data Mining. 2020.
>
> [2] Chen, Jialin, et al. "D4Explainer: In-distribution Explanations of Graph Neural Network via Discrete Denoising Diffusion." Thirty-seventh Conference on Neural Information Processing Systems. 2023.
>
> [3] Xiaoqi Wang and Han Wei Shen. GNNInterpreter: A probabilistic generative model-level explanation for graph neural networks. In The Eleventh International Conference on Learning Representations, 2023. URL https://openreview.net/forum?id=rqq6Dh8t4d.

---

### Meta-Review · Area_Chair_UWV2 · 2023-12-20

**Metareview:**

All reviewers found that this paper focuses on an important problem and that using decision boundaries as a tool for GNN explanation is innovative and elegant. They unanimously recommend Accept.

**Justification For Why Not Higher Score:**

There are opportunities for improvements such as more diverse tasks and more analysis.

**Justification For Why Not Lower Score:**

This paper focus on a timely problem with an elegant solution together with solid empirical study.

---

### Decision · Program_Chairs · 2024-01-16

Accept (poster)